# MULTI-DOMAIN IMAGE GENERATION AND TRANSLATION WITH IDENTIFIABILITY GUARANTEES

**Shaoan Xie**[1], **Lingjing Kong**[1], **Mingming Gong**[3,2], and **Kun Zhang**[1,2]

[1] Carnegie Mellon University
[2] Mohamed bin Zayed University of Artificial Intelligence
[3] The University of Melbourne
shaoan@cmu.edu, lingjingkong@cmu.edu,
mingming.gong@unimelb.edu.au, kunz1@cmu.edu

## ABSTRACT

Multi-domain image generation and unpaired image-to-to-image translation are two important and related computer vision problems. The common technique for the two tasks is the learning of a joint distribution from multiple marginal distributions. However, it is well known that there can be infinitely many joint distributions that can derive the same marginals. Hence, it is necessary to formulate suitable constraints to address this highly ill-posed problem. Inspired by the recent advances in nonlinear Independent Component Analysis (ICA) theory, we propose a new method to learn the joint distribution from the marginals by enforcing a specific type of minimal change across domains. We report one of the first results connecting multi-domain generative models to identifiability and shows why identifiability is essential and how to achieve it theoretically and practically. We apply our method to five multi-domain image generation and six image-to-image translation tasks. The superior performance of our model supports our theory and demonstrates the effectiveness of our method. The training code are available at https://github.com/Mid-Push/i-stylegan.

## 1 INTRODUCTION

Multi-domain image generation and unpaired image-to-image translation are two important and closely related problems in computer vision and machine learning. They have many promising applications such as domain adaptation (Liu & Tuzel, 2016; Hoffman et al., 2018; Murez et al., 2018; Wang & Jiang, 2019) and medical analysis (Armanious et al., 2019; 2020; Kong et al., 2021). As shown in Fig. 1, multi-domain image generation takes as input the random noise $\epsilon$ and domain label **u** and the task aims to generate image tuples where the images in the tuple share the same content, e.g., different facial expressions of the same people. The second task takes as input an image in one domain and target domain label **u** and aims to generate another image which is in the target domain but share the same content of input, e.g., the output image has the same identity but different facial expression from the input image.

Both tasks can be viewed as instantiations of joint distribution learning problem. A joint distribution of multi-domain images is a probability density function that gives a density value to each joint occurrence of images in different domains such as images of the same people with different facial expressions. Once the joint distribution is learned, it can be used to generate meaningful tuples (the first task) and translate an input image into another domain without content distortion (the second task). If the correspondence across domains is given (e.g., the identity), we can apply supervised approaches to learn the joint distribution easily. However, collecting corresponding data across domains can be prohibitively expensive . For example,

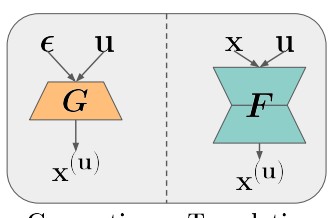

Figure 1: Two tasks.

collecting different facial expressions of same person may need controlled experiments.

By contrast, collecting image domains without correspondence can be relatively cheap, e.g., facial expressions of different people can be easily accessed online (once permission is granted). Therefore, we consider the problem of unsupervised joint distribution learning $P(\mathbf{x}^{(1)}, \mathbf{x}^{(2)}, ..., \mathbf{x}^{(d)})$ where we are only given multiple marginal distributions in different yet related domains $\{P(\mathbf{x}^{(i)})\}_{i=1}^{n}$, where $d$ is the number of domains and $\mathbf{x}^{(i)}$ denotes the images in domain $i$. However, there can be infinitely many joint distributions that can produce the same marginals (Lindvall, 2002). For example, we can apply conditional GANs (Mirza & Osindero, 2014; Odena et al., 2017; Miyato & Koyama, 2018; Brock et al., 2018) to match the marginal distributions in each domain and they learn joint distributions implicitly. We can sample a tuple $\langle x^{(0)}, ..., x^{(d)} \rangle$ from its learned joint distribution using the same random noise and different domain labels. However, there is no guarantee that the learned joint distribution in conditional GANs are optimal and it may drop the correspondence in the generated tuples (e.g., Fig.5(b)). To tackle this problem, CoGAN (Liu & Tuzel, 2016) proposes to use different generators for each domain and share weights of high level layers in different generators. JointGAN (Pu et al., 2018) proposes to factorize the joint distribution (e.g., $P(\mathbf{x}^{(1)}, \mathbf{x}^{(2)}) = P(\mathbf{x}^{(1)})P(\mathbf{x}^{(2)}|\mathbf{x}^{(1)})$) and learn one marginal and one conditional distribution with cycle consistency (Zhu et al., 2017).

While existing approaches (e.g., Liu et al. (2017); Pu et al. (2018) add constraints for the purpose of removing unwanted joint distributions, they are not guaranteed to find the true joint distribution. If we are unsure about the learned joint distribution, we cannot guarantee that the generated tuples are meaningful. Learning the true joint distribution seems to be impossible since we only have access to marginal distributions and domain label and we do not have access to the latent content and style variables. Fortunately, recent advances in nonlinear Independent Component Analysis (ICA) theory (Hyvarinen & Morioka, 2016; 2017; Hyvarinen et al., 2019; Khemakhem et al., 2020b;b; Von Kügelgen et al., 2021; Kong et al., 2022) show that deep nonlinear latent variable models are identifiable with auxiliary variables (e.g., domain label in our case), meaning that we can recover the latent variables (e.g., content and style in our case) up to some component-wise transformation. Inspired by these advances, we propose a new method to learn the true joint distribution from the marginals by enforcing a specific type of minimal changes across domains. Specifically, we assume that the influence of domain information (i.e., the underlying changes across domains) is minimal in the data generation process, e.g., only facial expression changes and no hair style changes is allowed. To achieve the minimal changes, we inject the domain information through a component-wise strictly increasing transformation instead arbitrary complex transformation. In addition, we assume the number of the underlying components affected by the domain information (and thus with changing distributions) is minimal. Then we show that if the influence of domain information is minimal, the true joint distribution can be recovered from the marginal distributions. Afterwards, we can use the learned joint distribution to sample meaningful tuples and translate input images into another domain without content distortion. Our method can be applied to datasets where the content across domains are aligned and most existing datasets in multi-domain image generation and translation satisfy this requirement, e.g., the facial image dataset, animal face and digit images. Our method may not be as effective when the contents when some domains contain unaligned contents. We may need to pay more attention to the artworks-related dataset, since different painters focus on different objects in their painting (e.g., there are many fruit paintings by Cezanne and landscape paintings by Monet).

Our proposed method has several theoretical and practical contributions:

1. We provide the properties of an image generation process under which the true joint distribution can be recovered from marginal distributions.

2. In light of our theoretical results, we provide a practical method for multi-domain image generation. The proposed method can automatically determine the underlying dimension of the latent variables with changing distributions. Our method achieves promising results across five image generation tasks.

3. We propose to encourage the mapping function in image translation task to preserve the correspondence learned by our image generation model. The significant gain over the baseline methods on six image translation tasks demonstrate the effectiveness of our technique.

## 2 RELATED WORK

**Multi-domain image generation and translation** Generative Adversarial Network (GAN) (Goodfellow et al., 2014) performs adversarial training between the generator and the discriminator. At the end, the distribution of sample generated by the generator is matched to the real data distribution. Conditional GAN (Mirza & Osindero, 2014) is a variant of GAN that incorporates additional information and have been widely applied for class-conditional image generation (Brock et al., 2018). CoGAN (Liu & Tuzel, 2016) learns the joint distribution by sharing the higher layers of two generators. JointGAN (Pu et al., 2018) proposes to factorize the joint distribution and learn marginal and conditionals and regularize the conditionals with cycle consistency. RegCGAN (Mao & Li, 2018) penalizes the distance between the features of the synthesized pairs. Methods by Liu & Tuzel (2016); Pu et al. (2018); Mao & Li (2018) are shown to be effective on some tasks but they do not have identifiability guarantees that they are recovering the true joint distribution (we provide a formal definition of identifiability later). Unpaired image-to-image translation relies on additional assumptions to address the task and cycle consistency (Zhu et al., 2017) is arguably most widely used. However, it has been argued that cycle consistency may not be enough to learn a good mapping (Alami Mejjati et al., 2018; Xie et al., 2022). Therefore, we propose a new regularization with generated tuples to help find a good mapping. We provide more related works in appendix C.

**Nonlinear ICA** Nonlinear ICA aims to recover independent latents from the data that are generated by nonlinear invertible transformations from the underlying independent variables (Hyvarinen & Morioka, 2016; 2017; Hyvarinen et al., 2019). Recent works have shown that the true latent variables may be identifiable given additional information (Khemakhem et al., 2020a; Gresele et al., 2020; Locatello et al., 2020; Shu et al., 2019; Zimmermann et al., 2021; Hälvä & Hyvarinen, 2020; Klindt et al., 2020a)). Khemakhem et al. (2020a) prove that the latent variable is identifiable if the prior distribution is conditionally factorized. Von Kügelgen et al. (2021) show that the latent content variable is block-identifiable given two views of same image, which is not applicable in our case since we do not have paired data. In the next section, we will show how our formulated problem is related to some variant of nonlinear ICA and how we can establish the identifiability of the changes across domains.

## 3 UNSUPERVISED JOINT DISTRIBUTION LEARNING

In this section, we first provide the formulations of our conditional generative model and some additional conditioins, under which the true joint distribution is identifiable. Then we provide a practical implementation based on conditional GAN to achieve unsupervised multi-domain image generation. Finally, we propose a novel regularization technique to improve unpaired image-to-image translation.

### 3.1 IDENTIFIABILITY OF THE JOINT DISTRIBUTION

Given $d$ marginal distributions $\{P_{\boldsymbol{\theta}}(\mathbf{x}^{(i)})\}_{i=1}^d$ derived from the true but unknown joint distribution $P_{\boldsymbol{\theta}}(\mathbf{x}^{(1)}, ..., \mathbf{x}^{(d)})$ where $\boldsymbol{\theta} \in \Theta$ is a vector of parameters, we need to recover the true joint distribution to generate meaningful image tuples or translating images without content distribution. To recover the true joint distribution, it needs to be identifiable. Formally, the true joint distribution is *identifiable* if following hold:

$$\forall \boldsymbol{\theta}' : \{P_{\boldsymbol{\theta}}(\mathbf{x}^{(i)}) = P_{\boldsymbol{\theta}'}(\mathbf{x}^{(i)})\}_{i=1}^n \Leftrightarrow P_{\boldsymbol{\theta}}(\mathbf{x}^{(1)}, \mathbf{x}^{(2)}, ..., \mathbf{x}^{(d)}) = P_{\boldsymbol{\theta}'}(\mathbf{x}^{(1)}, \mathbf{x}^{(2)}, ..., \mathbf{x}^{(d)}). \quad (1)$$

That is, if another model with parameter $\boldsymbol{\theta}' \in \Theta$ matches the marginal distributions in each domain, then this would imply that the true joint distribution is also matched perfectly by the model with parameter $\boldsymbol{\theta}'$. This is a well-known ill-posed problem: the joint distribution is not identifiable from marginal distributions without further assumptions. Motivated by the multiple domain image generation and translation problems, we define the following data generation process using a latent variable model (we provide the graphical model in appendix D):

$$P_{\boldsymbol{\theta}}(\mathbf{x}^{(1)}, ..., \mathbf{x}^{(d)}) = \int \cdots \int \prod_{i=1}^d P_{\boldsymbol{\theta}}(\mathbf{x}^{(i)}|\mathbf{z}_c, \mathbf{z}_s^{(i)}) P_{\boldsymbol{\theta}}(\mathbf{z}_c) \prod_{i=1}^d P_{\boldsymbol{\theta}}(\mathbf{z}_s^{(i)}) d\mathbf{z}_c d\mathbf{z}_s^{(1)} \ldots d\mathbf{z}_s^{(d)}, \quad (2)$$

where $\mathbf{z}_c$ is the common content and $\mathbf{z}_s^{(i)}$ is the style in the $i$-th domain. For instance, in expression generation, $\mathbf{z}_c$ represents the human identity information, and $\mathbf{z}_s$ represents the expression. Then

a data point sampled from the joint distribution of our model would be one person with different expressions. In practice, we may not have sample from the joint distribution; instead, we only have samples from the marginal distributions of $P_{\boldsymbol{\theta}}(\mathbf{x}^{(1)}, ..., \mathbf{x}^{(d)})$:

$$P_{\boldsymbol{\theta}}(\mathbf{x}^{(\mathbf{u})}) = \int P_{\boldsymbol{\theta}}(\mathbf{x}|\mathbf{z}_c, \mathbf{z}_s)P_{\boldsymbol{\theta}}(\mathbf{z}_c)P_{\boldsymbol{\theta}}(\mathbf{z}_s|\mathbf{u})d\mathbf{z}_c d\mathbf{z}_s, \tag{3}$$

where $\mathbf{u}$ is the domain label and $P_{\boldsymbol{\theta}}(\mathbf{z}_s|\mathbf{u} = i) = P_{\boldsymbol{\theta}}(\mathbf{z}_s^{(i)})$. To recover the true joint distribution, we first recover the true content $\mathbf{z}_c$ and true style $\mathbf{z}_s$ up to some transformation and then we establish the identifiability of the true joint distribtion.

Recovering $\mathbf{z}_c$ and $\mathbf{z}_s$ seems to be impossible since we only have the observations $\{\mathbf{x}^{(\mathbf{u})}\}$ and domain label $\mathbf{u}$. Fortunately, recent advances on nonlinear ICA theory have shown that deep non-linear latent variable models are identifiable given auxiliary variable (e.g., domain label in our case) (Khemakhem et al., 2020b;a; Von Kügelgen et al., 2021; Kong et al., 2022). Specifically, we define

$$P_{\boldsymbol{\theta}}(\mathbf{z}_s|\mathbf{u}) = \frac{P(\tilde{\mathbf{z}}_s)}{|\mathbf{J}_{f_{\mathbf{u}}}|}; \quad \mathbf{z}_s = f_{\mathbf{u}}(\tilde{\mathbf{z}}_s), \tilde{\mathbf{z}}_s \sim P(\tilde{\mathbf{z}}_s), \tag{4}$$

where $f_{\mathbf{u}}$ is domain-specific *component-wise strictly increasing* transformation (i.e., in each domain, each dimension of $\mathbf{z}_s$ is a strictly increasing function of the corresponding dimension of $\tilde{\mathbf{z}}_s$) and $|\mathbf{J}_{f_{\mathbf{u}}}|$ is the absolute value of determinant of the Jacobian matrix of function $f_{\mathbf{u}}$. $P(\tilde{\mathbf{z}}_s)$ is a prior distribution and we define it as $\mathcal{N}(\mathbf{0}, \mathbf{I})$. Then we also define:

$$P_{\boldsymbol{\theta}}(\mathbf{z}_c) = P(\mathbf{z}_c); P_{\boldsymbol{\theta}}(\mathbf{x}|\mathbf{z}_c, \mathbf{z}_s) = \delta(x - g(\mathbf{z}_c, \mathbf{z}_s)), \tag{5}$$

where $g$ is the true generation function, $\delta(.)$ is the Dirac delta function and $P(\mathbf{z}_c)$ is a prior distribution and we also define it as $\mathcal{N}(\mathbf{0}, \mathbf{I})$ in our model.

We now have $\boldsymbol{\theta} = (f_{\mathbf{u}}, g)$ and we can train a model with parameter $\boldsymbol{\theta}' = (\hat{f}_{\mathbf{u}}, \hat{g})$ to match the marginal distributions $P_{\boldsymbol{\theta}}(\mathbf{x}^{(i)})$. As for the true variables $(\mathbf{z}_c, \mathbf{z}_s, \mathbf{x}^{(\mathbf{u})})$, we also have the learned ones $(\hat{\mathbf{z}}_c, \hat{\mathbf{z}}_s, \hat{\mathbf{x}}^{(\mathbf{u})})$. Now we show that the true content $\mathbf{z}_c$ and style $\mathbf{z}_s$ are identifiable up to some transformations. We first define $\mathbf{z} = [\mathbf{z}_c; \mathbf{z}_s]$, $n = \dim(\mathbf{z})$, $n_s = \dim(\mathbf{z}_s)$, $n_c = \dim(\mathbf{z}_c)$, $\mathcal{Z} \subseteq \mathbb{R}^n$ is the domain of latent variable $\mathbf{z}$, $\mathcal{Z}_c \subseteq \mathbb{R}^{n_c}$ is the domain of latent variable $\mathbf{z}_c$, $\mathcal{Z}_s \subseteq \mathbb{R}^{n_s}$ is the domain of latent variable $\mathbf{z}_s$, $\mathcal{U}$ is the support of the distribution of $\mathbf{u}$. We say $\mathbf{z}_s$ is component-wise identifiable if there exists a component-wise invertible transformation $h_s$ s.t. $\hat{\mathbf{z}}_s = h_s(\mathbf{z}_s)$ for the recovered $\hat{\mathbf{z}}_s$. The content $\mathbf{z}_c$ is block-wise identifiable means that there exists an invertible function $h_c$ s.t. $\hat{\mathbf{z}}_c = h_c(\mathbf{z}_c)$ for the recovered $\hat{\mathbf{z}}_c$.

**Lemma 3.1.** *If the underlying data generation process is consistent with (3,4,5) and following assumptions hold:*

- *A1 (Smooth and Positive Density): The probability density function of latent variables is smooth (the second order derivative of the log density exists) and positive i.e. $P(\mathbf{z}|\mathbf{u})$ is smooth and $P(\mathbf{z}|\mathbf{u}) > 0$ for all $\mathbf{z} \in \mathcal{Z}$ and $\mathbf{u} \in \mathcal{U}$.*

- *A2 (Conditional independence): Conditioned on $\mathbf{u}$, the components of $\mathbf{z}$ are mutually independent, which implies $P(\mathbf{z}|\mathbf{u}) = \prod_i^n P(\mathbf{z}_i|\mathbf{u})$.*

- *A3 (Linear independence): For any $\mathbf{z}_s \in \mathcal{Z}_s \subseteq \mathbb{R}^{n_s}$, there exist $2n_s + 1$ values of $\mathbf{u}$, i.e., $\mathbf{u}_j$ with $j = 0, 1, ..., 2n_s$, such that the $2n_s$ vectors $\mathbf{w}(\mathbf{z}_s, \mathbf{u}_j) - \mathbf{w}(\mathbf{z}_s, \mathbf{u}_0)$ with $j = 1, ..., 2n_s$, are linearly independent, where vector $\mathbf{w}(\mathbf{s}, \mathbf{u})$ is defined as follows:*

$$\mathbf{w}(\mathbf{z}_s, \mathbf{u}) = \left( \frac{\partial q_{n_c+1}(\mathbf{z}_{n_c+1}, \mathbf{u})}{\partial \mathbf{z}_{n_c+1}}, \dots, \frac{\partial q_n(\mathbf{z}_n, \mathbf{u})}{\partial \mathbf{z}_n}, \frac{\partial^2 q_{n_c+1}(\mathbf{z}_{n_c+1}, \mathbf{u})}{\partial \mathbf{z}_{n_c+1}^2}, \dots, \frac{\partial^2 q_n(\mathbf{z}_n, \mathbf{u})}{\partial \mathbf{z}_n^2} \right),$$

  *where $q_i(\mathbf{z}_i, \mathbf{u}) = \log P(\mathbf{z}_i|\mathbf{u})$.*

- *A4 (Domain Variability): For any set $A_{\mathbf{z}} \subseteq \mathcal{Z}$ with the following two properties: (1) $A_{\mathbf{z}}$ has nonzero probability measure, i.e. $\mathbb{P}(\{\mathbf{z} \in A_{\mathbf{z}}\}|\{\mathbf{u} = \mathbf{u}'\}) > 0$ for any $\mathbf{u}' \in \mathcal{U}$. (2) $A_{\mathbf{z}}$ cannot be expressed as $B_{\mathbf{z}_c} \times \mathcal{Z}_s$ for any set $B_{\mathbf{z}_c} \subset \mathcal{Z}_c$.*

  *$\exists \mathbf{u}_1, \mathbf{u}_2 \in \mathcal{U}$, such that $\int_{\mathbf{z} \in A_{\mathbf{z}}} P(\mathbf{z}|\mathbf{u}_1)d\mathbf{z} \neq \int_{\mathbf{z} \in A_{\mathbf{z}}} P(\mathbf{z}|\mathbf{u}_2)d\mathbf{z}$.*

*by matching the marginal distributions $\{P_{\boldsymbol{\theta}}(\mathbf{x}^{(i)})\}_{i=1}^{d}$ of each domain, the component-wise identifiability of the style $\mathbf{z}_s$ is ensured, $\mathbf{z}_c$ is block-wise identifiable.*

We provide the proof and conditions in the supplementary. Intuitively, assumptions $A3$ and $A4$ require that the distribution $P(\mathbf{z}|\mathbf{u})$ varies sufficiently across domains such that the we have sufficient contrastive information to achieve the identifiability. A3 requires that the changes in the probability density functions are complex enough such that they don't always lie within $(2n_s + 1)$-dimensional subspace. Kong et al. (2022) provide a similar result in the context of domain adaptation but it requires the dimensions of input and output of the generating function $g$ to be the same. If we use GAN, the dimension of input noise and output image are obviously different, so our results are more general.

After recovering the true content $\mathbf{z}_c$ with $\hat{\mathbf{z}}_c$ and true style $\mathbf{z}_s$ with $\hat{\mathbf{z}}_s$, we now proceed to address the joint distribution identifiability problem. A main challenge is the indeterminacy of the recovered content $\mathbf{z}_c$ and style $\mathbf{z}_s$ caused by the unknown transformation $h_c$ and $h_s$. Fortunately, we show that this indeterminacy can be removed and the recovered joint distribution is the true one.

**Theorem 3.2.** *If 1) the underlying data generation process is consistent with (3,4,5), 2) assumptions A1-A4 hold, and 3) the marginal distributions are matched, i.e., $P_{\boldsymbol{\theta}'}(\mathbf{x}^{(i)}) = P_{\boldsymbol{\theta}}(\mathbf{x}^{(i)})$ for any domain $i \in [d]$, then the true joint distribution is identical to that produced by the model with parameter $\boldsymbol{\theta}'$, i.e., $P_{\boldsymbol{\theta}'}(\mathbf{x}^{(1)}, \mathbf{x}^{(2)}, ..., \mathbf{x}^{(d)}) = P_{\boldsymbol{\theta}}(\mathbf{x}^{(1)}, \mathbf{x}^{(2)}, ..., \mathbf{x}^{(d)})$.*

The proof is provided in appendix F. Unlike existing approaches, this theorem provides a guarantee that we are able to recover the true joint distribution under some conditions. In other words, the tuples generated by the learned generative model with parameter $\boldsymbol{\theta}'$ can also be viewed being sampled from the true generative model. As a consequence, elements in the generated tuple $\langle x^{(1)}, ..., x^{(d)} \rangle$ have different styles but share the same content. We can apply this model to address the challenging multi-domain image generation task directly. As for unpaired image-to-image translation, we will show that we can also use the generated tuples to help improve the performance of image translation.

## 3.2 A PRACTICAL IMPLEMENTATION OF MULTI-DOMAIN IMAGE GENERATION

In this section, we provide a practical implementation of our conditional generative model and we can use it to achieve multi-domain image generation. Given images from $d$ domains, we would like to train a conditional GAN model such that the generated tuples $\langle G(\boldsymbol{\epsilon}, 1), ..., G(\boldsymbol{\epsilon}, d) \rangle$ share the content where $G$ is the generator in GAN and $\boldsymbol{\epsilon}$ is the random Gaussian noise and $\mathbf{u} \sim \{1, .., d\}$.

In order to generate meaningful tuples, we build our conditional GAN following (3,4,5). Given random noise $\boldsymbol{\epsilon} \sim \mathcal{N}(\mathbf{0}, \mathbf{I})$ and domain label $\mathbf{u}$, a naive way would be splitting the noise into two parts, i.e., $\boldsymbol{\epsilon} = [\hat{\mathbf{z}}_c; \tilde{\mathbf{z}}_s]$ and apply component-wise strictly increasing transformation $\hat{f}_{\mathbf{u}}$ on $\tilde{\mathbf{z}}_s$. But the problem is that the true latent variables $\mathbf{z}_c, \mathbf{z}_s$ are only identifiable when the dimensions match, i.e., $\dim(\hat{\mathbf{z}}_c) = n_c = \dim(\mathbf{z}_c)$ and $\dim(\hat{\mathbf{z}}_s) = n_s = \dim(\mathbf{z}_s)$. We usually do not have access to the true dimension $n_c$ and $n_s$. To address this problem, we have to assume the total dimension is matched, i.e., $n_s + n_c = \dim(\hat{\mathbf{z}}_c) + \dim(\hat{\mathbf{z}}_s)$. Then we only need to determine $n_s$. We may treat $n_s$ as a hyper-parameter and sweep the possible values. But it can be expensive since we have to determine the values for each dataset. To address this issue, we propose a simple way to allow the network to learn the optimal dimension automatically with an additional regularization term. The whole computation flow is shown in Fig. 2. Specifically, we apply component-wise strictly increasing transformation $\hat{f}_{\mathbf{u}}$ to all components of the input noise $\boldsymbol{\epsilon}$. Then we multiply it with a trainable mask $m$ and add it back to the input noise:

$$\hat{\mathbf{z}} = \boldsymbol{\epsilon} + m \odot \hat{f}_{\mathbf{u}}(\boldsymbol{\epsilon}), \quad G(\boldsymbol{\epsilon}, \mathbf{u}) = \hat{g}(\hat{\mathbf{z}}_c, \hat{\mathbf{z}}_s) \tag{6}$$

where the output $\hat{\mathbf{z}}$ is the input to the generator $G$. The mask $m$ is of the same dimension as noise $\boldsymbol{\epsilon}$ and the elements are in the range $[0, +\infty)$. For any element $i$, if $m_i = 0$, $\hat{\mathbf{z}}_i$ only contains information from the shared $\boldsymbol{\epsilon}$ and it belongs to $\hat{\mathbf{z}}_c$. Otherwise, $\hat{\mathbf{z}}_i$ contains the domain information $\mathbf{u}$ and it belongs to $\hat{\mathbf{z}}_s$. In order to encourage the network determine the optimal dimension automatically, we apply $L_1$ loss on the mask,

$$\mathcal{L}_{\text{sparsity}} = \|m\|_1. \tag{7}$$

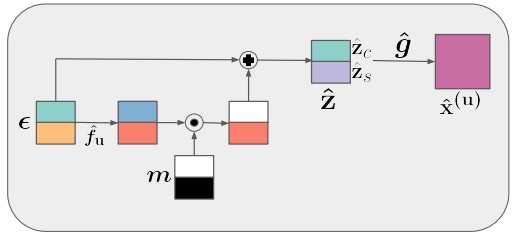

Figure 2: The computation flow of our model for multi-domain image generation. $\epsilon$ denotes the input noise, $\mathbf{u}$ is domain index, $\hat{f}_{\mathbf{u}}$ is the domain-specific component-wise strictly increasing transformation, $m$ is the trainable mask, $\hat{\mathbf{z}}_c$ is the content which is shared across domains while $\hat{\mathbf{z}}_s$ is the style which changes across domains and $\hat{\mathbf{x}}_{\mathbf{u}}$ is the output image in domain $\mathbf{u}$.

Finally, we can proceed as normal conditional GAN method. We introduce an conditional discriminator $D$ and training objective is:

$$\mathcal{L}_{\text{gan}} = \mathbb{E}[\log(D(\mathbf{x}, \mathbf{u}))] + \mathbb{E}[\log(1 - D(G(\epsilon, \mathbf{u}), \mathbf{u}))]. \tag{8}$$

Our full objective for multi-domain image generation is

$$\mathcal{L}_{\text{generation}} = \mathcal{L}_{\text{gan}} + \lambda \mathcal{L}_{\text{sparsity}}, \tag{9}$$

where $\lambda$ controls the influence of domain $\mathbf{u}$. We found that $\lambda = 0.1$ works well across all datasets and is more stable than tuning the dimension $n_s$ in section 4.1.2.

### 3.3 APPLICATION: UNPAIRED IMAGE-TO-IMAGE TRANSLATION

Unpaired image-to-image translation aims to map input images to a target domain while preserving important content information. This task can also be formulated as a joint distribution learning problem (Liu et al., 2017). For example, given two domains $\mathbf{x}^{(u_0)}$ and $\mathbf{x}^{(u_1)}$, image translation aims to learn a reasonable conditional distribution $P_{\boldsymbol{\theta}'}(\mathbf{x}^{(u_1)}|\mathbf{x}^{(u_0)})$ through a mapping function $F$, which should be close to the true conditional $P_{\boldsymbol{\theta}}(\mathbf{x}^{(u_1)}|\mathbf{x}^{(u_0)})$. Therefore, we can use our multi-domain image generation model to help find a proper conditional distribution. The differences between two tasks are visualized in Figure. 1.

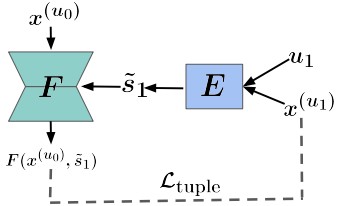

Figure 3: Our regularization.

We employ StarGAN-v2 (Choi et al., 2020) as our backbone method and other methods are also applicable. StarGAN-v2 mainly consists of: the mapping function $F$, style encoder $E$. The training loss is $\mathcal{L}_{\text{stargan}} = \mathcal{L}_{\text{adv}} + \mathcal{L}_{\text{cyc}} + \mathcal{L}_{\text{sty}} - \lambda_{div}\mathcal{L}_{div}$, where $\mathcal{L}_{adv}$, $\mathcal{L}_{cyc}$, $\mathcal{L}_{sty}$ and $\mathcal{L}_{div}$ are used for *distribution matching*, *cycle consistency*, *style reconstruction* and *generation diversity*, respectively. For more details, we refer readers to the original paper or our appendix J.1.

After matching the marginal distributions, $G(\epsilon, \mathbf{u})$ allows us to sample meaningful tuples by changing the value of domain label $\mathbf{u}$, e.g., $\langle x^{(u_0)} = G(\epsilon, u_0), x^{(u_1)} = G(\epsilon, u_1)\rangle$. Then we can use them to further regularize the mapping function $F$ as

$$\mathcal{L}_{\text{tuple}} = \mathbb{E}\|F(x^{(u_0)}, \tilde{s}_1) - x^{(u_1)}\|_1, \tag{10}$$

where $\tilde{s}_1 = E(x^{(u_1)}, u_1)$ is the style code of domain $u_1$. Through $\mathcal{L}_{\text{tuple}}$, we are encouraging the mapping function $F$ to reconstruct the corresponding image $G(\epsilon, u_1)$ from the input image $G(\epsilon, u_0)$. It means that the mapping function $F$ is trained to preserve the correspondence between images of the generated tuples. Since we are able to recover the true joint distribution, $\mathcal{L}_{\text{tuple}}$ encourage the mapping function to produce the true conditional distribution, i.e., $P_{\theta}(\mathbf{x}^{(u_1)}|\mathbf{x}^{(u_0)})$.

Our full objective for unpaired image translation is $\mathcal{L}_{\text{translation}} = \mathcal{L}_{\text{stargan}} + \lambda_{\text{tuple}}\mathcal{L}_{\text{tuple}}$, where $\lambda_{\text{tuple}}$ is the hyper-parameter to control the influence of our propose tuple regularization.

# 4 EXPERIMENTS

In this section, we first present results and analysis on multi-domain image generation task. Then we provide the results on unpaired image translation.

## 4.1 MULTI-DOMAIN IMAGE GENERATION

### 4.1.1 EXPERIMENT SETUP

**Implementation** We build our method based on the official pytorch implementation of StyleGAN2-ADA (Karras et al., 2020a) and the hyper-parameters are selected automatically by the code. We choose the deep sigmoid flow (DSF) (Huang et al., 2018a) to implement the domain transformation $f_{\mathbf{u}}$ (Huang et al., 2018a) because DSF is designed to be component-wise strictly increasing. We use the embedding of domain label to generate pseudo-parameters for the flow. We only introduce one hyper-parameter: $\lambda$ to control the sparsity of the mask. We set $\lambda = 0.1$ for all experiments.

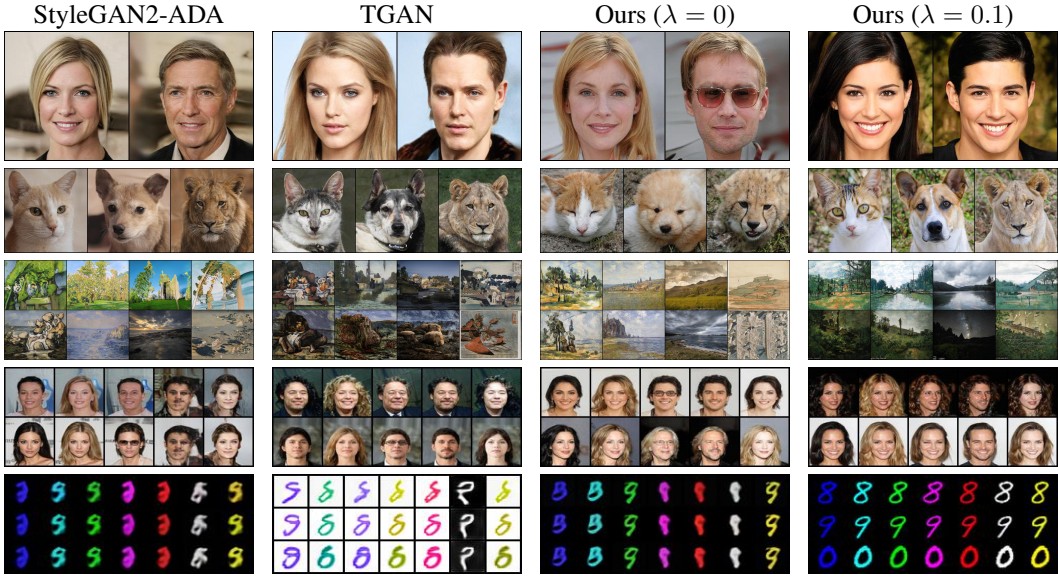

Figure 4: Samples of multi-domain image generation on the CELEBA-HQ, AFHQ, ArtPhoto, CelebA5 and MNIST7. We provide more samples and methods in appendix G.2. Each row of the method shares the same input noise $\epsilon$. We observe that there are unnecessary changes between the images (e.g., the added sun-glasses in the first row, the different poses of animals of StyleGAN2-ADA in second row) without regularization.

| Dataset | Metrics | StyleGAN2-ADA | TGAN | CoGAN | Ours ($\lambda = 0$) | Ours ($\lambda = 0.1$) |
|---------|---------|---------------|------|-------|---------------------|------------------------|
| CELBEA-HQ | FID ↓ | 4.97 | 5.97 | 5.17 | 5.51 | **4.87** |
| | DIPD ↓ | 0.61 | 0.60 | **0.58** | 0.60 | **0.58** |
| AFHQ | FID ↓ | 6.16 | 7.15 | 6.52 | 8.58 | **4.36** |
| | DIPD ↓ | 1.12 | **1.01** | 1.07 | 1.13 | 1.02 |
| ArtPhoto | FID ↓ | 10.13 | 9.72 | 10.47 | 15.65 | **9.71** |
| | DIPD ↓ | 1.56 | 1.63 | 1.57 | 1.65 | **1.49** |
| CelebA5 | FID ↓ | 9.82 | 2.77 | 6.11 | 3.86 | **2.69** |
| | DIPD | 0.40 | 0.25 | 0.38 | 0.26 | **0.23** |
| MNIST7 | FID ↓ | 95.9 | 210.8 | 103.0 | 83.4 | **1.43** |
| | Joint-FID ↓ | 160.70 | 277.00 | 152.22 | 144.90 | **9.73** |

Table 1: Results of multi-domain image generation on five datasets.

**Datasets** We use five datasets to evaluate our method: CELEBA-HQ (Choi et al., 2020) contains female and male faces domains; AFHQ (Choi et al., 2020) contains 3 domains: cat, dog and wild

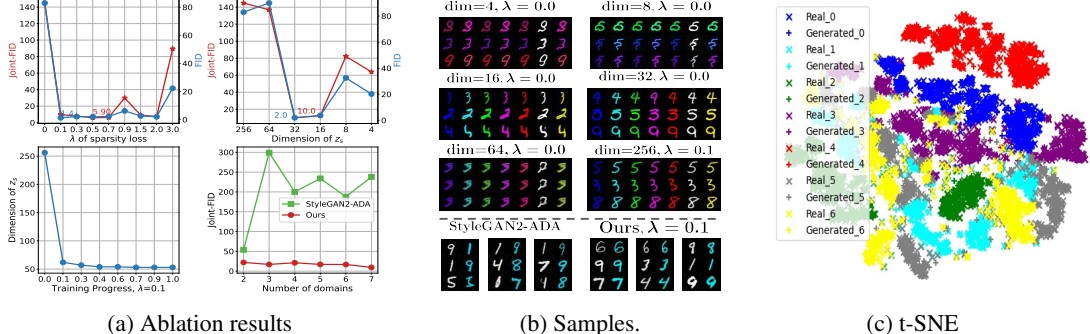

(a) Ablation results         (b) Samples.         (c) t-SNE

Figure 5: Experiments on MNIST7 dataset. We observe that our proposed sparsity constraint enables network select the optimal dimension of changing components automatically (bottom left in (a)) and is more stable than tuning the dimension manually (top left and right in the (a), examples in (b)). We visualize the real and generated samples with t-SNE in (c). Different colors denote different domains. We can find that the generated samples cover all classes in each domain. We provide t-SNE comparisons in the Fig. 9.

life; ArtPhoto contains 4 domains: Cezanne, Monet, Photo and Ukiyoe; CelebA5 contains 5 domains: Black Hair, Blonde Hair, Eyeglasses, Mustache and Pale Skin; MNIST7 contains 7 domains: blue, cyan, green, purple, red, white and yellow MNIST digits. More information are in the appendix G.1.

**Evaluation Metrics**. We evaluate our method using the Frechet inception distance (FID), which is a widely used metric for distribution divergence between the generated images and the real images. lower FID is better. As for the first four datasets, there is no pair data. So, we use the domain-invariant perceptual distance (DIPD) to measure the semantic correspondence (Liu et al., 2019). DIPD computes the distance between two instance-normalized Conv5 features of VGG network. As for MNIST7, we have ground truth tuples, so we first compute inception features of images in the tuple, concatenate the features and reduce the dimension by variance thresholding. Then we can compute the frechet distance between the features of ground truth and generated tuples. We name it Joint-FID as it computes the divergence between joint distributions.

**Baselines**. Since our method is built on StyleGAN2-ADA (Karras et al., 2020a), we also run StyleGAN2-ADA to verify the effectiveness of our introduced modules. We also run the most recent conditional GAN model - TGAN (Shahbazi et al., 2022). TGAN observes that the result of conditional generation of StyleGAN2-ADA is unsatisfactory when the number of classes are large. Therefore it starts as the unconditional StyeleGAN2-ADA and then gradually transits to the conditional training. We reimplement CoGAN (Liu & Tuzel, 2016) based on the StyleGAN2-ADA architecture. We let the generators share all synthesis blocks except the last one.

### 4.1.2 RESULTS AND ANALYSIS

**Comparison with Baselines** We present the quantitative results in Table 3 and show samples generated by different methods in Fig. 4. Images of each row are generated by the same Gaussian noise with different values of domain label **u**. Our method achieves best FID on all datasets. The significant improvement over StyleGAN2-ADA highlight the importance to consider the correspondence information across domains. Our method also achieves lowest DIPD values on three datasets. The low values of DIPD demonstrates that images in our method share more content than other methods. In other words, our method is able to learn a proper joint distribution given only marginals.

**The proposed sparsity loss reduces the influence of domain stably and help us avoid expensive hyper-parameter sweeping**. As we mentioned in section 3.2, a naive way to further reduce the influence of domain label **u** is to reducing the dimension of $\mathbf{z}_s$ directly. But the results can be sensitive to the dimension $n_s$. To verify this point, we sweep the $n_s$ in [256, 64,32,16,8,4] (and set $\lambda = 0$) and show the results in Fig. 5(a) (top two rows). We observe that our sparsity constraint works well if $\lambda$ is in the range [0,1], which is a common range for hyper-parameter. However, tuning $n_s$ works well when $n_s = 16, 32$. We plot the valid number of $\hat{\mathbf{z}}_s$ by checking the value of the trainable mask $m$ in Fig. 5(a) (bottom left), it learns to decrease $n_s$ to around 50, which is close to the optimal values by

tuning $n_s$. The samples in Fig. 5(b) also suggests that the generated tuples with sparsity loss share more content (when dim=32, it confuses between digit 3 and 5, second row).

**The identifiability when the number of domain is small** The identifiability of the latent variables requires $2n_s + 1$ domains as shown in section 3.1. Therefore, it would be interesting to see what happens when the number of domains is small. We have shown that our method achieves the best result across different real world datasets in Table 3. Now, we choose MNIST7 dataset and decrease the number of domains used in the training. We show the results in Fig. 5 (a) (bottom right). We only compute the Joint-FID of the first two domains to ensure the results are comparable. We can observe a clear trend that the Joint-FID is improving as we increase the number of domains, which supports our theory. We observe that the StyleGAN2-ADA achieves Joint-FID of 54 and our method achieves 22 when there are only 2 domains. We also show the generated examples in Fig.5 (b), last panel. StyleGAN2-ADA totally drops the correspondence between images in the generated tuples while our method is able to preserve the correspondence.

## 4.2 APPLICATION: IMAGE-TO-IMAGE TRANSLATION

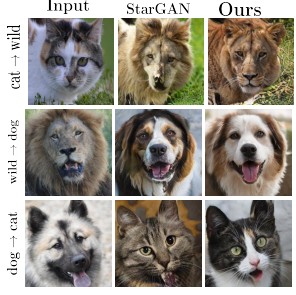

|  | latent | | reference | |
| **Method** | **FID**↓ | **LPIPS** ↑ | **FID** ↓ | **LPIPS** ↑ |
| MUNIT (Huang et al., 2018b) | 41.5 | 0.511 | 223.9 | 0.199 |
| DRIT (Lee et al., 2018) | 95.6 | 0.326 | 114.8 | 0.156 |
| MSGAN (Mao & Li, 2018) | 61.4 | 0.517 | 69.8 | 0.375 |
| StarGAN-v2 (Choi et al., 2020) | 16.2 | 0.450 | 19.8 | 0.432 |
| SmoothGAN (Liu et al., 2021) | 51.9 | 0.418 | 52.1 | 0.342 |
| LETIT (Zhao & Chen, 2021) | 15.9 | - | - | - |
| StyleDis (Kim et al., 2022) | - | - | 14.7 | - |
| Ours | **11.1** | **0.518** | **13.4** | **0.481** |

Table 2: Average results on 6 tasks

Figure 6: Samples of image translation.

### 4.2.1 RESULTS

We present the experiment setup in the appendix J. We present the average results of 6 pairwise image-to-image translation tasks in Table 2. We can observe significant improvements over StarGAN-V2 in both latent and reference based tasks. The significant improvement of our method indicates that the generated tuples can help match the distribution as well as improving the diversity of the outputs. We present more samples in the appendix J.

## 5 LIMITATION AND FUTURE WORK

Our model is trained to learn the true joint distribution. However, an essential assumption is that the content $\mathbf{z}_c$ across domains are aligned. For example, the female and male domains share the same human content. For more complex data where the content are not aligned, our assumption may be violated. For example, we find that Cezanne domain contains many fruit paintings while other domains contains landscape in the ArtPhoto dataset. Although our method has

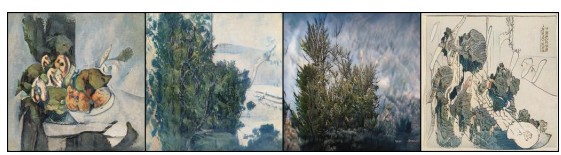

Figure 7: Failure case on ArtPhoto dataset, columns are Cezanne, Monet, Photo and Ukyioe

achieved best quality as well as semantic correspondence compared to other methods, we still observe some failure cases. As shown in Figure. 7, to match the distribution of Cezanne domain, the generator learns to generate the fruit painting (first column). Although we can observe that our method still works on other three domains (the rest columns), we can observe content mismatch due to the unaligned training dataset. Our proposed architecture may over-constrain the influence of domain label in this misalignment case. We may also resort to some other quantitative measures of the influence to help address this challenging problem and we leave it as future work.

ACKNOWLEDGEMENT

We thank the anonymous reviewers for their devoted time and constructive feedbacks, which are really helpful to improve the quality of this paper.

This project was partially supported by the National Institutes of Health (NIH) under Contract R01HL159805, by the NSF-Convergence Accelerator Track-D award 2134901, by a grant from Apple Inc., a grant from KDDI Research Inc, and generous gifts from Salesforce Inc., Microsoft Research, and Amazon Research. MG was supported by ARC DE210101624.

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

## A    Ethical Statement

Generative models such as GAN used in our paper enable various applications and our model facilitates the technique and may reduce the manual labeling cost. Unfortunately, as revealed by previous reports, it becomes easier to manipulate image data, such as the deepfakes. In addition, the generative models may reveal the training data information. How to address these negative impacts still remains an important problem. All datasets we used are publicly available.

## B    Discussion about our model and the conditional GANs

Conditional GANs usually feed the concatenation of the input noise $\epsilon$ and domain label $\mathbf{u}$ into the generator $G$. By contrast, we inject the domain influence to the noise $\epsilon$ through a component-wise strictly increasing transformation $\hat{f}_{\mathbf{u}}$. From a theoretic perspective, our architecture allows us to recover the true joint distribution as shown in section 3.1. From an empirical perspective, we would like to reduce the influence of domain information in our generative model. To generate images in each domain, the generator needs to utilize the input domain label $\mathbf{u}$. However, the influence of $\mathbf{u}$ can be very large if no constraint is applied. If the influence of domain variable is unnecessarily large, the generator focuses on the domain variable $\mathbf{u}$ and pays little attention to the content variable $\mathbf{z}_c$. As a consequence, the generated images in the tuple may lose correspondence (e.g., the generated animals have different poses on AFHQ dataset, Fig. 4). An extreme case would be that the content is totally ignored by the generator. So the generator still outputs images in different domains (because the input domain labels are different ) but all images in the same domain look the same (because the content variable $z_c$ is ignored) (e.g., the generated digits look the same on MNIST7 dataset, Fig. 4). By contrast, we first use the simple transformation $f_{\mathbf{u}}$ to inject the domain information into $\epsilon$ rather than concatenating it with $\epsilon$. Secondly, the sparsity loss $\mathcal{L}_{\text{sparsity}}$ reduces the number of components affected by the domain label $\mathbf{u}$. Therefore, there are will only be necessary changes between the images in the generated tuples.

## C    Related Work

**Multi-domain Image Generation** Conditional GAN aims to maximize the influence of conditioning variable to improve the diversity of the generated samples (Gong et al., 2019; Brock et al., 2018; Miyato & Koyama, 2018; Odena et al., 2017; Kang et al., 2021; Kang & Park, 2020; Tseng et al., 2021; Karras et al., 2020a; Miyato et al., 2018; Zhang et al., 2019a; Brock et al., 2018; Wu et al., 2019; Zhang et al., 2019b; Zhao et al., 2020). Our task is also related to the conditional GAN problem if we regard the domain label as the conditioning information. A major difference is that conditional GAN mostly focusing on enlarging the influence of the condition class label and generate diverse images across classes (Kang et al., 2021) while our method is trying to reduce the influence of the condition domain label and generate meaningful tuples across domains.

**Unpaired Image-to-Image Translation** Image-to-image translation can also be viewed as a joint distribution learning problem between the source and the target image domain. With pair data, pix2pix (Isola et al., 2017) applies conditional GAN to match the distribution of the target domain and penalize the distance of generated image to the ground truth image. Unfortunately, paired data is usually difficult to collect. Therefore, additional assumptions are made to address the unsupervised task, such as the cycle consistency (Zhu et al., 2017; Kim et al., 2017; Yi et al., 2017; Choi et al., 2018; 2020; Liu et al., 2021; Kim et al., 2022), shared latent space (Liu et al., 2017; Huang et al., 2018b; Lee et al., 2018; Yu et al., 2019; Liu et al., 2018), relationship preservation(Park et al., 2020; Han et al., 2021; Wang et al., 2021; Cao et al., 2019; Xu et al., 2022), density changing Xie et al. (2022), importance reweighting Xie et al. (2021). We build our method based on StarGAN-v2 (Choi et al., 2020), which relies on cycle consistency to preserve content.

## D    The Graphical Model

To address the ill-posed problem, we assume the generation process follows following graphical model:

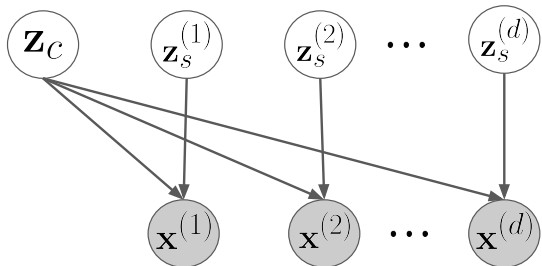

Figure 8: The graphical model of our method. $\mathbf{z}_c$ is the shared content and $\mathbf{z}^{(i)}$ is the style from domain $i$ and $\mathbf{x}^{(i)}$ is the observation: data (images) in domain $i$.

# E    PROOF OF THE IDENTIFIABILITY OF LATENT VARIABLES

We show that the true content $\mathbf{z}_c$ and style $\mathbf{z}_s$ are identifiable in Lemma. 3.1.

## E.1    THE IDENTIFIABILITY OF THE STYLE

We first provide the proof of the identifiability of the style variable $\mathbf{z}_s$.

*Proof.* As we have matched marginal distributions $P_{\boldsymbol{\theta}'}(\hat{\mathbf{x}}|\mathbf{u}) = P_{\boldsymbol{\theta}}(\mathbf{x}|\mathbf{u})$, we have $P(\hat{g}(\hat{\mathbf{z}})|\mathbf{u}) = P(g(\mathbf{z})|\mathbf{u})$ since $\hat{\mathbf{x}} = \hat{g}(\hat{\mathbf{z}})$, $\mathbf{x} = g(\mathbf{z})$. Then we can apply same transformation $\hat{g}^{-1}$ to the variables $\hat{g}(\hat{\mathbf{z}})$ and $g(\mathbf{z})$, which results in $P(\hat{g}^{-1} \circ \hat{g}(\hat{\mathbf{z}})|\mathbf{u}) = P(\hat{g}^{-1} \circ g(\mathbf{z})|\mathbf{u}) \Rightarrow P(\hat{\mathbf{z}}|\mathbf{u}) = P(\hat{g}^{-1} \circ g(\mathbf{z})|\mathbf{u})$. We define $h = g^{-1} \circ \hat{g}$, then $h^{-1} = \hat{g}^{-1} \circ g$, we have

$$P(\hat{\mathbf{z}}|\mathbf{u}) = P(h^{-1}(\mathbf{z})|\mathbf{u}), \tag{11}$$

which suggests that $h$ is the indeterminacy between the recovered latent variable $\hat{\mathbf{z}}$ and the true latent $\mathbf{z}$. It is worth noting that we don't need to assume that the dimension of $\mathbf{z}$ and $\mathbf{x}$ are same in order to compute the determinant of $g^{-1}$.

We further define $q_i^{\mathbf{u}} = \log P(\mathbf{z}_i|\mathbf{u})$ and $\hat{q}_i^{\mathbf{u}} = \log P(\hat{\mathbf{z}}_i|\mathbf{u})$. With the conditional independence assumption in A2, we have

$$P(\mathbf{z}|\mathbf{u}) = \prod_{i=1}^{n} P(\mathbf{z}_i|\mathbf{u}) \Leftrightarrow \log P(\mathbf{z}|\mathbf{u}) = \sum_{i=1}^{n} q_i^{\mathbf{u}}$$

$$P(\hat{\mathbf{z}}|\mathbf{u}) = \prod_{i=1}^{n} P(\hat{\mathbf{z}}_i|\mathbf{u}) \Leftrightarrow \log P(\hat{\mathbf{z}}|\mathbf{u}) = \sum_{i=1}^{n} \hat{q}_i^{\mathbf{u}}$$

According to the change of variable rule, we can transform equation. 11 into

$$P(\hat{\mathbf{z}}|\mathbf{u}) = \frac{1}{|\mathbf{J}_{h^{-1}}|} P(\mathbf{z}|\mathbf{u}) \cdot \iff \sum_{i=1}^{n} \hat{q}_i^{\mathbf{u}} = \sum_{i=1}^{n} q_i^{\mathbf{u}} + |\mathbf{J}_h| \tag{12}$$

where $\mathbf{J}_{h^{-1}}$ is the absolute value of the determinant of the Jacobian matrix of $h$. Since $h$ is invertible and the input and output share the same number of dimension, we have $\frac{1}{|\mathbf{J}_{h^{-1}}|} = |\mathbf{J}_h| \neq 0$.

To simplify the notation, we define the following objects:

$$h'_{i,(k)} := \frac{\partial \mathbf{z}_i}{\partial \hat{\mathbf{z}}_k}, \qquad h''_{i,(k,q)} := \frac{\partial^2 \mathbf{z}_i}{\partial \hat{\mathbf{z}}_k \partial \hat{\mathbf{z}}_q};$$

$$\eta'_i(\mathbf{z}_i, \mathbf{u}) := \frac{\partial q_i^{\mathbf{u}}}{\partial \mathbf{z}_i}, \qquad \eta''_i(\mathbf{z}_i, \mathbf{u}) := \frac{\partial^2 q_i^{\mathbf{u}}}{(\partial \mathbf{z}_i)^2}.$$

Differentiating Equation 12 twice w.r.t. $\hat{\mathbf{z}}_k$ and $\hat{\mathbf{z}}_q$ where $k, q \in [n]$ and $k \neq q$ yields

$$\sum_{i=1}^{n} \left( \eta_i''(\mathbf{z}_i, \mathbf{u}) \cdot h_{i,(k)}' h_{i,(q)}' + \eta_i'(\mathbf{z}_i, \mathbf{u}) \cdot h_{i,(k,q)}'' \right) + \frac{\partial^2 \log |\mathbf{J}_h|}{\partial \hat{\mathbf{z}}_k \partial \hat{\mathbf{z}}_q} = 0. \tag{13}$$

Therefore, there are $2n_s + 1$ equations corresponding to $\mathbf{u} = \mathbf{u}_0, \ldots, \mathbf{u}_{2n_s}$ respectively. We subtract equations associated with $\mathbf{u}_1, \ldots, \mathbf{u}_{2n_s}$ with the equation of $\mathbf{u}_0$, and we obtain the following $2n_s$ equations:

$$\sum_{i=n_c+1}^{n} \left( (\eta_i''(\mathbf{z}_i, \mathbf{u}_j) - \eta_i''(\mathbf{z}_i, \mathbf{u}_0)) \cdot h_{i,(k)}' h_{i,(q)}' + (\eta_i'(\mathbf{z}_i, \mathbf{u}_j) - \eta_i'(\mathbf{z}_i, \mathbf{u}_0)) \cdot h_{i,(k,q)}'' \right) = 0, \tag{14}$$

where $j = 1, \ldots 2n_s$. Due to the invariance of $\tilde{z}_c$, we have $P(\mathbf{z}_c) = P(\mathbf{z}_c|\mathbf{u})$. Thus, we have $\eta_i''(\mathbf{z}_i, \mathbf{u}_j) = \eta_i''(\mathbf{z}_i, \mathbf{u}_{j'})$ and $\eta_i'(\mathbf{z}_i, \mathbf{u}_j) = \eta_i'(\mathbf{z}_i, \mathbf{u}_{j'}), \forall j, j'$. Hence only the style components $i = n_c + 1, \ldots, n$ remain in the summation of each equation.

Under the linear independence condition in Assumption A3, the linear system is a $2n_s \times 2n_s$ invertible. Therefore, the only solution is $h_{i,(k)}' h_{i,(q)}' = 0$ and $h_{i,(k,q)}'' = 0$ for $i = n_c + 1, \ldots, n$ and $k, q \in [n], k \neq q$.

As $h(\cdot)$ is smooth over $\mathcal{Z}$, its Jacobian can be written as:

$$\mathbf{J}_h = \begin{bmatrix} \mathbf{A} := \frac{\partial \mathbf{z}_c}{\partial \hat{\mathbf{z}}_c} & \mathbf{B} := \frac{\partial \mathbf{z}_c}{\partial \hat{\mathbf{z}}_s} \\ \mathbf{C} := \frac{\partial \mathbf{z}_s}{\partial \hat{\mathbf{z}}_c} & \mathbf{D} := \frac{\partial \mathbf{z}_s}{\partial \hat{\mathbf{z}}_s}, \end{bmatrix} \tag{15}$$

Note that $h_{i,(k)}' h_{i,(q)}' = 0$ implies that for each $i = n_c + 1, \ldots, n$, $h_{i,(k)}' \neq 0$ for at most one element $k \in [n]$. Therefore, there is only at most one non-zero entry in each row indexed by $i = n_c + 1, \ldots, n$ in the Jacobian matrix $\mathbf{J}_{h^{-1}}$. Further, the invertibility of $h(\cdot)$ necessitates $\mathbf{J}_{h^{-1}}$ to be full-rank which implies that there is exactly one non-zero component in each row of matrices $\mathbf{C}$ and $\mathbf{D}$.

Since for every $i \in \{n_c + 1, \ldots, n\}$, $\hat{\mathbf{z}}_i$ has changing distributions over $\mathbf{u}$ and all $\hat{\mathbf{z}}_k$'s for $i \in \{1, \ldots, n_c\}$ (i.e. $\hat{\mathbf{z}}_c$) have invariant distributions over $\mathbf{u}$, we can deduce that $\mathbf{C} = \mathbf{0}$ and the only non-zero entry $\frac{\hat{z}_i}{z_k}$ must reside in $\mathbf{D}$ with $k \in \{n_c + 1, \ldots, n\}$. Therefore, for each estimated variable in the changing part $\hat{z}_i, i \in \{n_c + 1, \ldots, n\}$, there exists one true variable in the changing part $\mathbf{z}_k, k \in \{n_c + 1, \ldots, n\}$ such that $\hat{z}_i = h_i'(\mathbf{z}_k)$. Further, because $\mathbf{J}_{h^{-1}}$ is of full-rank ($h(\cdot)$ being invertible) and $\mathbf{C}$ is a zero matrix, $\mathbf{D}$ must be of full-rank, which implies that $h_i'(\cdot)$ is invertible for each $i \in \{n_c + 1, \ldots, n\}$. Thus, the changing components $\mathbf{z}_s$ are identified up to component-wise invertible transformations.

$\square$

## E.2 THE IDENTIFIABILITY OF THE CONTENT VARIABLE

Now we provide the proof for the block-identifiability of the content variable $\mathbf{z}_c$.

*Proof.* The proof is presented in four steps as follows.

**Step 1.** Due to assumption of the data generating process of the learned model, we can obtain the independence between the generating process $\hat{\mathbf{z}}_c$ and $\mathbf{u}$. Thus, it follows that for any $A_{\mathbf{z}_c} \subseteq \mathcal{Z}_c$,

$$\{\hat{g}_{1:n_c}^{-1}(\hat{\mathbf{x}}) \in A_{\mathbf{z}_c}\}|\{\mathbf{u} = \mathbf{u}_1\} = \{\hat{g}_{1:n_c}^{-1}(\hat{\mathbf{x}}) \in A_{\mathbf{z}_c}\}|\{\mathbf{u} = \mathbf{u}_2\}, \quad \forall \mathbf{u}_1, \mathbf{u}_2 \in \mathcal{U}$$

$$\Longleftrightarrow$$

$$\{\hat{\mathbf{x}} \in (\hat{g}_{1:n_c}^{-1})^{-1}(A_{\mathbf{z}_c})\}|\{\mathbf{u} = \mathbf{u}_1\} = \{\hat{\mathbf{x}} \in (\hat{g}_{1:n_c}^{-1})^{-1}(A_{\mathbf{z}_c})\}|\{\mathbf{u} = \mathbf{u}_2\}, \quad \forall \mathbf{u}_1, \mathbf{u}_2 \in \mathcal{U} \tag{16}$$

where $\hat{g}_{1:n_c}^{-1} : \mathcal{X} \to \mathcal{Z}_c$ denotes the estimated transformation from the observation to the content variable and $(\hat{g}_{1:n_c}^{-1})^{-1}(A_{\mathbf{z}_c}) \subseteq \mathcal{X}$ is the pre-image set of $A_{\mathbf{z}_c}$.

On account of the matching assumption, we are able to extend Equation 16 as follows:

$$\{\mathbf{x} \in (\hat{g}_{1:n_c}^{-1})^{-1}(A_{\mathbf{z}_c})\}|\{\mathbf{u} = \mathbf{u}_1\} = \{\mathbf{x} \in (\hat{g}_{1:n_c}^{-1})^{-1}(A_{\mathbf{z}_c})\}|\{\mathbf{u} = \mathbf{u}_2\}$$

$$\Longleftrightarrow$$

$$\{\hat{g}_{1:n_c}^{-1}(\mathbf{x}) \in A_{\mathbf{z}_c}\}|\{\mathbf{u} = \mathbf{u}_1\} = \{\hat{g}_{1:n_c}^{-1}(\mathbf{x}) \in A_{\mathbf{z}_c}\}|\{\mathbf{u} = \mathbf{u}_2\}. \tag{17}$$

Because $g$ and $\hat{g}$ are smooth and injective, there exists a smooth and injective $h = \hat{g}^{-1} \circ g : \mathcal{Z} \to \mathcal{Z}$. Expressing $\hat{g}^{-1} = h \circ g^{-1}$ and $h_c(\cdot) := h_{1:n_c}(\cdot) : \mathcal{Z} \to \mathcal{Z}_c$ in Equation 17 yields

$$\{h_c(\mathbf{z}) \in A_{\mathbf{z}_c}\}|\{\mathbf{u} = \mathbf{u}_1\} = \{h_c(\mathbf{z}) \in A_{\mathbf{z}_c}\}|\{\mathbf{u} = \mathbf{u}_2\},$$

$$\Longleftrightarrow$$

$$\{\mathbf{z} \in h_c^{-1}(A_{\mathbf{z}_c})\}|\{\mathbf{u} = \mathbf{u}_1\} = \{\mathbf{z} \in h_c^{-1}(A_{\mathbf{z}_c})\}|\{\mathbf{u} = \mathbf{u}_2\},$$

$$\Longleftrightarrow$$

$$\int_{\mathbf{z} \in h_c^{-1}(A_{\mathbf{z}_c})} p_{\mathbf{z}|\mathbf{u}}(\mathbf{z}|\mathbf{u}_1)\, d\mathbf{z} = \int_{\mathbf{z} \in h_c^{-1}(A_{\mathbf{z}_c})} p_{\mathbf{z}|\mathbf{u}}(\mathbf{z}|\mathbf{u}_2)\, d\mathbf{z}, \tag{18}$$

where $h_c^{-1}(A_{\mathbf{z}_c}) = \{\mathbf{z} \in \mathcal{Z} : h_c(\mathbf{z}) \in A_{\mathbf{z}_c}\}$ is the pre-image of $A_{\mathbf{z}_c}$, i.e. those latent variables containing content variables in $A_{\mathbf{z}_c}$ after the indeterminacy transformation $h$.

Based on the generating process, we can re-write Equation 18 as follows: $\forall A_{\mathbf{z}_c} \subseteq \mathcal{Z}_c$,

$$\int_{[\mathbf{z}_c^\top, \mathbf{z}_s^\top]^\top \in h_c^{-1}(A_{\mathbf{z}_c})} p_{\mathbf{z}_c}(\mathbf{z_c}) \left( p_{\mathbf{z}_s|\mathbf{u}}(\mathbf{z_s}|\mathbf{u}_1) - p_{\mathbf{z}_s|\mathbf{u}}(\mathbf{z_s}|\mathbf{u}_2) \right) d\mathbf{z}_s d\mathbf{z}_c = 0. \tag{19}$$

**Step 2.** In this step, we prove that $\mathbf{z}_c := h_c([\mathbf{z}_c^\top, \mathbf{z}_s^\top]^\top)$ does not depend on $\mathbf{z}_s$. To this end, we first develop an equivalent statement (i.e. Statement 3 below) and prove it subsequently. This enables us to leverage the full-supported density function assumption to avert technical issues.

- Statement 1: $h_c([\mathbf{z}_c^\top, \mathbf{z}_s^\top]^\top)$ does not depend on $\mathbf{z}_s$.

- Statement 2: $\forall \mathbf{z}_c \in \mathcal{Z}_c$, it follows that $h_c^{-1}(\mathbf{z}_c) = B_{\mathbf{z}_c} \times \mathcal{Z}_s$ where $B_{\mathbf{z}_c} \neq \emptyset$ and $B_{\mathbf{z}_c} \subseteq \mathcal{Z}_c$.

- Statement 3: $\forall \mathbf{z}_c \in \mathcal{Z}_c, r \in \mathbf{R}^+$, it follows that $h_c^{-1}(\mathcal{B}_r(\mathbf{z}_c)) = B_{\mathbf{z}_c}^+ \times \mathcal{Z}_s$ where $\mathcal{B}_r(\mathbf{z}_c) := \{\mathbf{z}_c' \in \mathcal{Z}_c : ||\mathbf{z}_c' - \mathbf{z}_c||^2 < r\}$, $\bar{B}_{\mathbf{z}_c} \neq \emptyset$, and $B_{\mathbf{z}_c}^+ \subseteq \mathcal{Z}_c$.

Statement 2 is a mathematical formulation of Statement 1. Statement 3 generalizes singletons $\mathbf{z}_c$ in Statement 2 to open, non-empty balls $\mathbf{B}_r(\mathbf{z}_c)$. Later, we show that under this generalization, $\bar{B}_{\mathbf{z}_c}$ is necessarily of probability measure greater than $0$. With this, we can proceed to show its contraction to Equation 19.

Leveraging the continuity of $h_c(\cdot)$, we show the equivalence between Statement 2 and Statement 3 as follows. We first show that Statement 2 implies Statement 3. $\forall \mathbf{z}_c \in \mathcal{Z}_c, r \in \mathbb{R}^+$, $h_c^{-1}((\mathcal{B}_r(\mathbf{z}_c))) = \cup_{\mathbf{z}_c' \in \mathcal{B}_r(\mathbf{z}_c)} h_c^{-1}(\mathbf{z}_c')$. Statement 2 indicates that every participating sets in the union satisfies $h_c^{-1}(\mathbf{z}_c') = B_{\mathbf{z}_c}' \times \mathcal{Z}_s$, thus the union $h_c^{-1}((\mathcal{B}_r(\mathbf{z}_c)))$ also satisfies this property, which is Statement 3.

Then, we show that Statement 3 implies Statement 2 by contradiction. Suppose that Statement 2 is false, then $\exists \hat{\mathbf{z}}_c \in \mathcal{Z}_c$ such that there exist $\hat{\mathbf{z}}_c^B \in \{\mathbf{z}_{1:n_c} : \mathbf{z} \in h_c^{-1}(\hat{\mathbf{z}}_c)\}$ and $\hat{\mathbf{z}}_s^B \in \mathcal{Z}_s$ resulting in $h_c(\hat{\mathbf{z}}^B) \neq \hat{\mathbf{z}}_c$ where $\hat{\mathbf{z}}^B = [(\hat{\mathbf{z}}_c^B)^\top, (\hat{\mathbf{z}}_s^B)^\top]^\top$. As $h_c(\cdot)$ is continuous, there exists $\hat{r} \in \mathbb{R}^+$ such that $h_c(\hat{\mathbf{z}}^B) \notin \mathcal{B}_{\hat{r}}(\hat{\mathbf{z}}_c)$. That is, $\hat{\mathbf{z}}^B \notin h_c^{-1}(\mathcal{B}_{\hat{r}}(\hat{\mathbf{z}}_c))$. On the other hand, Statement 3 suggests that $h_c^{-1}(\mathcal{B}_{\hat{r}}(\hat{\mathbf{z}}_c)) = \hat{B}_{\mathbf{z}_c} \times \mathcal{Z}_s$. By definition of $\hat{\mathbf{z}}^B$, it is clear that $\hat{\mathbf{z}}_{1:n_c}^B \in \hat{B}_{\mathbf{z}_c}$. However, the fact that $\hat{\mathbf{z}}^B \notin h_c^{-1}(\mathcal{B}_{\hat{r}}(\hat{\mathbf{z}}_c))$ contradicts Statement 3. Therefore, Statement 2 is true under the premise of Statement 3. We have shown that Statement 3 implies Statement 2. Consequently, Statement 2 and Statement 3 are equivalent, and therefore proving Statement 3 suffices to show Statement 1.

**Step 3.** In this step, we prove Statement 3 by contradiction. Intuitively, we show that if $h_c(\cdot)$ depended on $\hat{z}_s$, the pre-image $h_c^{-1}(\mathcal{B}_r(\mathbf{z}_c))$ could be partitioned into two parts (i.e. $B_{\hat{\mathbf{z}}}^*$ and $h_c^{-1}(A_{\mathbf{z}_c}^*) \setminus B_{\hat{\mathbf{z}}}^*$ defined below). The dependency between $h_c(\cdot)$ and $\hat{z}_s$ is characterized by $B_{\hat{\mathbf{z}}}^*$, which

would not emerge otherwise. In contrast, $h_c^{-1}(A_{\mathbf{z}_c}^*) \setminus B_{\mathbf{z}}^*$ also exists when $h_c(\cdot)$ does not depend on $\hat{z}_s$. We evaluate the invariance relation Equation 19 and show that the integral over $h_c^{-1}(A_{\mathbf{z}_c}^*) \setminus B_{\mathbf{z}}^*$ (i.e. $T_1$) is always 0, however, the integral over $B_{\mathbf{z}}^*$ (i.e. $T_2$) is necessarily non-zero, which leads to the contraction with Equation 19 and thus shows the $h_c(\cdot)$ cannot depend on $\hat{z}_s$.

First, note that because $\mathcal{B}_r(\mathbf{z}_c)$ is open and $h_c(\cdot)$ is continuous, the pre-image $h_c^{-1}(\mathcal{B}_r(\mathbf{z}_c))$ is open. In addition, the continuity of $h(\cdot)$ and the matched observation distributions $\forall \mathbf{u}' \in \mathcal{U}, \; \{\mathbf{x} \in A_{\mathbf{x}}\}|\{\mathbf{u} = \mathbf{u}'\} = \{\hat{\mathbf{x}} \in A_{\mathbf{x}}\}|\{\mathbf{u} = \mathbf{u}'\}$ lead to $h(\cdot)$ being bijection as shown in (Klindt et al., 2020b),which implies that $h_c^{-1}(\mathcal{B}_r(\mathbf{z}_c))$ is non-empty. Hence, $h_c^{-1}(\mathcal{B}_r(\mathbf{z}_c))$ is both non-empty and open. Suppose that $\exists A_{\mathbf{z}_c}^* := \mathcal{B}_{r^*}(\mathbf{z}_c^*)$ where $\mathbf{z}_c^* \in \mathcal{Z}_c$, $r^* \in \mathbb{R}^+$, such that $B_{\mathbf{z}}^* := \{\mathbf{z} \in \mathcal{Z} : \mathbf{z} \in h_c^{-1}(A_{\mathbf{z}_c}^*), \{\mathbf{z}_{1:n_c}\} \times \mathcal{Z}_s \not\subseteq h_c^{-1}(A_{\mathbf{z}_c}^*)\} \neq \emptyset$. Intuitively, $B_{\mathbf{z}}^*$ contains the partition of the pre-image $h_c^{-1}(A_{\mathbf{z}_c}^*)$ that the style part $\mathbf{z}_{n_c+1:n}$ cannot take on any value in $\mathcal{Z}_s$. Only certain values of the style part were able to produce specific outputs of indeterminacy $h_c(\cdot)$. Clearly, this would suggest that $h_c(\cdot)$ depends on $\mathbf{z}_c$.

To show contraction with Equation 19, we evaluate the LHS of Equation 19 with such a $A_{\mathbf{z}_c}^*$:

$$
\int_{[\mathbf{z}_c^\top, \mathbf{z}_s^\top]^\top \in h_c^{-1}(A_{\mathbf{z}_c}^*)} p_{\mathbf{z}_c}(\mathbf{z_c}) \left( p_{\mathbf{z}_s|\mathbf{u}}(\mathbf{z_s}|\mathbf{u}_1) - p_{\mathbf{z}_s|\mathbf{u}}(\mathbf{z_s}|\mathbf{u}_2) \right) d\mathbf{z}_s d\mathbf{z}_c
$$

$$
= \underbrace{\int_{[\mathbf{z}_c^\top, \mathbf{z}_s^\top]^\top \in h_c^{-1}(A_{\mathbf{z}_c}^*) \setminus B_{\mathbf{z}}^*} p_{\mathbf{z}_c}(\mathbf{z_c}) \left( p_{\mathbf{z}_s|\mathbf{u}}(\mathbf{z_s}|\mathbf{u}_1) - p_{\mathbf{z}_s|\mathbf{u}}(\mathbf{z_s}|\mathbf{u}_2) \right) d\mathbf{z}_s d\mathbf{z}_c}_{T_1}
$$

$$
+ \underbrace{\int_{[\mathbf{z}_c^\top, \mathbf{z}_s^\top]^\top \in B_{\mathbf{z}}^*} p_{\mathbf{z}_c}(\mathbf{z_c}) \left( p_{\mathbf{z}_s|\mathbf{u}}(\mathbf{z_s}|\mathbf{u}_1) - p_{\mathbf{z}_s|\mathbf{u}}(\mathbf{z_s}|\mathbf{u}_2) \right) d\mathbf{z}_s d\mathbf{z}_c}_{T_2} \, .
$$

We first look at the value of $T_1$. When $h_c^{-1}(A_{\mathbf{z}_c}^*) \setminus B_{\mathbf{z}}^* = \emptyset$, $T_1$ evaluates to 0. Otherwise, by definition we can rewrite $h_c^{-1}(A_{\mathbf{z}_c}^*) \setminus B_{\mathbf{z}}^*$ as $C_{\mathbf{z}_c}^* \times \mathcal{Z}_s$ where $C_{\mathbf{z}_c}^* \neq \emptyset$ and $C_{\mathbf{z}_c}^* \subset \mathcal{Z}_c$. With this expression, it follows that

$$
\int_{[\mathbf{z}_c^\top, \mathbf{z}_s^\top]^\top \in C_{\mathbf{z}_c}^*} p_{\mathbf{z}_c}(\mathbf{z_c}) \left( p_{\mathbf{z}_s|\mathbf{u}}(\mathbf{z_s}|\mathbf{u}_1) - p_{\mathbf{z}_s|\mathbf{u}}(\mathbf{z_s}|\mathbf{u}_2) \right) d\mathbf{z}_s d\mathbf{z}_c
$$

$$
= \int_{\mathbf{z}_c \in C_{\mathbf{z}_c}^*} p_{\mathbf{z}_c}(\mathbf{z_c}) \int_{\mathbf{z}_s \in \mathcal{Z}_s} \left( p_{\mathbf{z}_s|\mathbf{u}}(\mathbf{z_s}|\mathbf{u}_1) - p_{\mathbf{z}_s|\mathbf{u}}(\mathbf{z_s}|\mathbf{u}_2) \right) d\mathbf{z}_s d\mathbf{z}_c
$$

$$
= \int_{\mathbf{z}_c \in C_{\mathbf{z}_c}^*} p_{\mathbf{z}_c}(\mathbf{z_c}) \left( 1 - 1 \right) d\mathbf{z}_c = 0.
$$

Therefore, in both cases $T_1$ evaluates to 0 for $A_{\mathbf{z}_c}^*$.

Now, we address $T_2$. As discussed above, $h_c^{-1}(A_{\mathbf{z}_c}^*)$ is open and non-empty. Because of the continuity of $h_c(\cdot)$, $\forall \mathbf{z}_B \in B_{\mathbf{z}}^*$, there exists $r(\mathbf{z}_B) \in \mathbb{R}^+$ such that $\mathcal{B}_{r(\mathbf{z}_B)}(\mathbf{z}_B) \subseteq B_{\mathbf{z}}^*$. As $p_{\mathbf{z}|\mathbf{u}}(\mathbf{z}|\mathbf{u}) > 0$ over $(\mathbf{z}, \mathbf{u})$, we have $\{\mathbf{z} \in B_{\mathbf{z}}^*\}|\{\mathbf{u} = \mathbf{u}'\} \geq \{\mathbf{z} \in \mathcal{B}_{r(\mathbf{z}_B)}(\mathbf{z}_B)|\{\mathbf{u} = \mathbf{u}'\}\} > 0$ for any $\mathbf{u}' \in \mathcal{U}$. Assumption A4 indicates that $\exists \mathbf{u}_1^*, \mathbf{u}_2^*$, such that

$$
T_2 := \int_{[\mathbf{z}_c^\top, \mathbf{z}_s^\top]^\top \in B_{\mathbf{z}}^*} p_{\mathbf{z}_c}(\mathbf{z_c}) \left( p_{\mathbf{z}_s|\mathbf{u}}(\mathbf{z_s}|\mathbf{u}_1^*) - p_{\mathbf{z}_s|\mathbf{u}}(\mathbf{z_s}|\mathbf{u}_2^*) \right) d\mathbf{z}_s d\mathbf{z}_c \neq 0.
$$

Therefore, for such $A_{\mathbf{z}_c}^*$, we would have $T_1 + T_2 \neq 0$ which leads to contradiction with Equation 19. We have proved by contradiction that Statement 3 is true and hence Statement 1 holds, that is, $h_c(\cdot)$ does not depend on the style variable $\mathbf{z}_s$.

**Step 4.** With the knowledge that $h_c(\cdot)$ does not depend on the style variable $\mathbf{z}_s$, we now show that there exists an invertible mapping between the true content variable $\mathbf{z}_c$ and the estimated version $\hat{\mathbf{z}}_c$.

As $h(\cdot)$ is smooth over $\mathcal{Z}$, its Jacobian can written as:

$$
\mathbf{J}_h = \left[ \begin{array}{cc} \mathbf{A} := \frac{\partial \hat{\mathbf{z}}_c}{\partial \mathbf{z}_c} & \mathbf{B} := \frac{\partial \hat{\mathbf{z}}_c}{\partial \mathbf{z}_s} \\ \hline \mathbf{C} := \frac{\partial \hat{\mathbf{z}}_s}{\partial \mathbf{z}_c} & \mathbf{D} := \frac{\partial \hat{\mathbf{z}}_s}{\partial \mathbf{z}_s}, \end{array} \right] \tag{20}
$$

where we use notation $\hat{\mathbf{z}}_c = h(\mathbf{z})_{1:n_c}$ and $\hat{\mathbf{z}}_s = h(\mathbf{z})_{n_c+1:n}$. As we have shown that $\hat{\mathbf{z}}_c$ does not depend on the style variable $\mathbf{z}_s$, it follows $\mathbf{B} = \mathbf{0}$. On the other hand, as $h(\cdot)$ is invertible over $\mathcal{Z}$, $\mathbf{J}_h$ is non-singular. Therefore, $\mathbf{A}$ must be non-singular due to $\mathbf{B} = \mathbf{0}$. Note that $\mathbf{A}$ is the Jacobian of the function $h'_c(\mathbf{z}_c) := h_c(\mathbf{z}) : \mathcal{Z}_c \to \mathcal{Z}_c$, which takes only the content part $\mathbf{z}_c$ of the input $\mathbf{z}$ into $h_c$. Also, note that the result of Theorem 3.1 implies that $\mathbf{C} = \mathbf{0}$. Together with the invertibility of $h$, we can conclude that $h'_c$ is invertible. Therefore, there exists an invertible function $h'_c$ between the estimated and the true content variables such that $\hat{\mathbf{z}}_c = h'_c(\mathbf{z}_c)$, which concludes the proof that $\mathbf{z}_c$ is block-identifiable via $\hat{g}^{-1}(\cdot)$. $\qquad\square$

# F    PROOF OF THE IDENTIFIABILITY OF TRUE JOINT DISTRIBUTION

In this section, we provide the proof of theorem 3.2.

*Proof.* **Two domains** We consider the case of two variables first, i.e., we prove that $P_{\boldsymbol{\theta}'}(\mathbf{x}^{(1)}, \mathbf{x}^{(2)}) = P_{\boldsymbol{\theta}}(\mathbf{x}^{(1)}, \mathbf{x}^{(2)})$. We factorize the joint distribution $P(\mathbf{x}^{(1)}, \mathbf{x}^{(2)}) = P(\mathbf{x}^{(1)})P(\mathbf{x}^{(2)}|\mathbf{x}^{(1)})$. Since we have matched the marginal distributions for each domain, we already have $P_{\boldsymbol{\theta}'}(\mathbf{x}^{(1)}) = P_{\boldsymbol{\theta}}(\mathbf{x}^{(1)})$. So we now only need to prove $P_{\boldsymbol{\theta}'}(\mathbf{x}^{(2)}|\mathbf{x}^{(1)}) = P_{\boldsymbol{\theta}}(\mathbf{x}^{(2)}|\mathbf{x}^{(1)})$.

Given $\mathbf{x} = g(\mathbf{z}_c, \mathbf{z}_s)$, we can identify the style and content through the inverse of the model, i.e., $\hat{\mathbf{z}}_c, \hat{\mathbf{z}}_s = \hat{g}^{-1}(\mathbf{x})$. According to lemma. 3.1, we have $h_c(\mathbf{z}_c) = \hat{\mathbf{z}}_c$ and $h_s(\mathbf{z}_s) = \hat{\mathbf{z}}_s$, where $h_c$ is an invertible transformation and $h_s$ is component-wise invertible transformation. Therefore, we have

$$g(\mathbf{z}_c, \mathbf{z}_s) = \hat{g}(\hat{\mathbf{z}}_c, \hat{\mathbf{z}}_s) = \hat{g}(h_c(\mathbf{z}_c), h_s(\mathbf{z}_s)) \tag{21}$$

For the ease of notation, we denote $\mathbf{c} = \mathbf{z}_c, \mathbf{s} = \mathbf{z}_s$. Given a pair of images $\langle x^{(1)}, x^{(2)} \rangle$ sampled from the true joint distribution, we have

$$x^{(1)} = g(c^{(1)}, s^{(1)}); \quad x^{(2)} = g(c^{(1)}, s^{(2)}); \tag{22}$$

Since we assume that the domain specific transformation is component-wise monotonic $f_{\mathbf{u}}$, $f_{\mathbf{u}}$ is component-wise invertible. We denote $f_1 = f_{\mathbf{u}=1}, f_2 = f_{\mathbf{u}=2}$, then we have

$$s^{(2)} = f_2 \circ f_1^{-1}(s^{(1)}) \quad s^{(1)} = f_2^{-1} \circ f_1(s^{(2)}) \tag{23}$$

Similarly, the learned model $\hat{f}_{\mathbf{u}}$ is also component-wise invertible, we also have

$$\hat{s}^{(2)} = \hat{f}_2 \circ \hat{f}_1^{-1}(\hat{s}^{(1)}). \tag{24}$$

Given $x^{(1)}$ in the first domain, we have

$$\hat{c}^{(1)} = h_c(c^{(1)}), \hat{s}^{(1)} = h_s(s^{(1)}); \quad x^{(1)} = g(c^{(1), s^{(1)}}) = \hat{g}(\hat{c}^{(1)}, \hat{s}^{(1)}). \tag{25}$$

Then we have

$$\begin{aligned} \hat{s}^{(2)} &= \hat{f}_2 \circ \hat{f}_1^{-1}(\hat{s}^{(1)}), \quad \text{by equality in 24} \\ &= \hat{f}_2 \circ \hat{f}_1^{-1}(h_s(s^{(1)})), \quad \text{by equality in 25} \\ &= \hat{f}_2 \circ \hat{f}_1^{-1}(h_s(f_1 \circ f_2^{-1}(s^{(2)}))), \quad \text{by equality in 23} \end{aligned} \tag{26}$$

We now prove that $\hat{s}^{(2)}$ is a function of the true $s^{(2)}$. According to our lemma, the function between $\hat{s}^{(2)}$ and $s^{(2)}$ can only be $h_s$. Therefore, we have $\hat{s}^{(2)} = h_s(s^{(2)})$. Then the output generated by our learned model with parameter $\boldsymbol{\theta}'$ is

$$\begin{aligned} \hat{x}^{(2)} &= \hat{g}(\hat{c}^{(1)}, \hat{s}^{(2)}) \\ &= \hat{g}(h_c(c^{(1)}), h_s(s^{(2)})) \\ &= g(c^{(1)}, s^{(2)}), \quad \text{by equality in 21} \\ &= x^{(2)} \end{aligned} \tag{27}$$

The results show that given a input $x^{(1)}$, our learned generative model with parameter $\boldsymbol{\theta}'$ outputs the same result as the true generative model with parameter $\boldsymbol{\theta}$. In other words, $P_{\boldsymbol{\theta}'}(\mathbf{x}^{(2)}|\mathbf{x}^{(1)}) = P_{\boldsymbol{\theta}}(\mathbf{x}^{(2)}|\mathbf{x}^{(1)})$.

**Generalization of the result with mathematical induction** So far, we have proved that $P_{\boldsymbol{\theta}'}(\mathbf{x}^{(1)}, \mathbf{x}^{(2)}) = P_{\boldsymbol{\theta}}(\mathbf{x}^{(1)}, \mathbf{x}^{(2)})$. Now we generalize the results into more domains with mathematical induction on the number of domains.

- **Base Case** When the number of domains is 2, we have proved that $P_{\boldsymbol{\theta}'}(\mathbf{x}^{(1)}, \mathbf{x}^{(2)}) = P_{\boldsymbol{\theta}}(\mathbf{x}^{(1)}, \mathbf{x}^{(2)})$.

- **Inductive Step**. Suppose the true joint distribution is identifiable when the number of domains is $d-1$, i.e., $P_{\boldsymbol{\theta}'}(\mathbf{x}^{(1)},...,\mathbf{x}^{(d-1)}) = P_{\boldsymbol{\theta}}(\mathbf{x}^{(1)},...,\mathbf{x}^{(d-1)})$, now we prove that the true joint distribution is still identifiable when the number of domains is $d$, i.e., $P_{\boldsymbol{\theta}'}(\mathbf{x}^{(1)},...,\mathbf{x}^{(d)}) = P_{\boldsymbol{\theta}}(\mathbf{x}^{(1)},...,\mathbf{x}^{(d)})$. In fact, we only need to prove $P_{\boldsymbol{\theta}'}(\mathbf{x})(\mathbf{x}^{(d)}|\mathbf{x}^{(1),...,\mathbf{x}^{(d-1)}}) = P_{\boldsymbol{\theta}'}(\mathbf{x})(\mathbf{x}^{(d)}|\mathbf{x}^{(1),...,\mathbf{x}^{(d-1)}})$ according to the induction assumption.

$$P(\mathbf{x}^{(d)}|\mathbf{x}^{(1),...,\mathbf{x}^{(d-1)}}) = P(\mathbf{x})(\mathbf{x}^{(d)}|c^{(1)}, s^{(1)}, c^{(1)}, s^{(2)}, ..., ), \text{ we apply the invertible function } g^{-1} \tag{28}$$

$$= P(\mathbf{x}^{(d)}|c^{(1)}, s^{(1)}, ..., s^{(d-1)}) \tag{29}$$

$$= P(\mathbf{x}^{(d)}|c^{(1)}, f_1^{-1}(s^{(1)}), ..., , f_{d-1}(s^{(d-1)})), \text{we apply the invertible transformations } f^{-1} \tag{30}$$

$$= P(\mathbf{x}^{(d)}|c^{(1)}, \tilde{s}, \tilde{s}, ..., \tilde{s}), \tag{31}$$

$$= P(\mathbf{x}^{(d)}|c^{(1)}, \tilde{s}) \tag{32}$$

$$= P(\mathbf{x}^{(d)}|c^{(1)}, f_1(\tilde{s})) \tag{33}$$

$$= P(\mathbf{x}^{(d)}|c^{(1)}, s^{(1)}) \tag{34}$$

We can find that it becomes the two domain case, i.e., we just need to prove $P_{\boldsymbol{\theta}'}(\mathbf{x}^{(d)}|\mathbf{x}^{(1)}) = P_{\boldsymbol{\theta}}(\mathbf{x}^{(d)}|\mathbf{x}^{(1)})$ , which already holds according to our results in two domains.

Therefore, the true joint distribution is identifiable. □

## G  MULTI-DOMAIN IMAGE GENERATION

### G.1  DATASET

We use five datasets to verify our model.

- CELEBA-HQ (Choi et al., 2020) contains 2 domain: female and male. We also use the training set to train our model. Female domain contains 17943 images and male domain contains 10057 images. We train the model at resolution $256\times256$.
- AFHQ (Choi et al., 2020) contains 3 domains: cat, dog and wild life (e.g., foxes, tigers and lions). We use the training set to train our conditional GAN model. Three domains contain 5153, 4739, 4738 images, respectively. We train the model at resolution $256 \times 256$.
- ArtPhoto contains 4 domains: Monet, Cezanne, Ukiyoe paintings and real photos (Zhu et al., 2017).
- CelebA5 contains 5 domains: Black Hair, Blonde Hair, Eyeglasses, Mustache and Pale Skin. They are subsets of domain CelebA (Liu et al., 2015). We train them at $64\times64$ resolution.
- MNIST7 contains 7 domains: blue, cyan, green, purple, red, white and yellow MNIST digits. We generate these digits using the training MNIST dataset (LeCun et al., 1998). We train the model at resolution $32\times32$.

### G.2  MORE RESULTS

We now provide more results of multi-domain image generation.

### G.3  T-SNE OF REAL AND GENERATED SAMPLE

## H  TWO VERSIONS OF MASK MECHANISM

In our main paper, we propose to use $z_s = \epsilon + m \odot f_u(\epsilon)$ to encourage the network select the optimal dimension of $n_s$ automatically. Another possible version would be $z_s^2 = (1-m_2) \odot \epsilon + m_2 \odot f_u(\epsilon)$.

| Dataset | Metrics | StyleGAN2-ADA | TGAN | CoGAN | Ours ($\lambda = 0$) | Ours ($\lambda = 0.1$) |
|---|---|---|---|---|---|---|
| CELBEA-HQ | FID $\downarrow$ | 4.97 | 5.97 | 5.17 | 5.51 | **4.87** |
| | DIPD $\downarrow$ | 0.61 | 0.60 | **0.58** | 0.60 | **0.58** |
| | Precison $\uparrow$ | 0.716 | 0.703 | 0.700 | 0.739 | 0.721 |
| | Recall $\uparrow$ | 0.454 | 0.382 | 0.418 | 0.391 | 0.445 |
| AFHQ | FID $\downarrow$ | 6.16 | 7.15 | 6.52 | 8.58 | **4.36** |
| | DIPD $\downarrow$ | 1.12 | **1.01** | 1.07 | 1.13 | 1.02 |
| | Precision $\uparrow$ | 0.757 | 0.710 | 0.745 | **0.769** | 0.706 |
| | Recall $\uparrow$ | 0.241 | 0.400 | 0.202 | 0.190 | **0.490** |
| ArtPhoto | FID $\downarrow$ | 10.13 | 9.72 | 10.47 | 15.65 | **9.71** |
| | DIPD $\downarrow$ | 1.56 | 1.63 | 1.57 | 1.65 | **1.49** |
| | Precision $\uparrow$ | 0.696 | 0.651 | 0.690 | **0.715** | 0.651 |
| | Recall $\uparrow$ | 0.281 | **0.373** | 0.298 | 0.106 | 0.359 |
| CelebA5 | FID $\downarrow$ | 9.82 | 2.77 | 6.11 | 3.86 | **2.69** |
| | DIPD $\downarrow$ | 0.40 | 0.25 | 0.38 | 0.26 | **0.23** |
| | Precision $\uparrow$ | **0.751** | 0.661 | 0.718 | 0.689 | 0.662 |
| | Recall $\uparrow$ | 0.323 | 0.510 | 0.181 | 0.452 | **0.537** |
| MNIST7 | FID $\downarrow$ | 95.9 | 210.8 | 103.0 | 83.4 | **1.43** |
| | Joint-FID $\downarrow$ | 160.70 | 277.00 | 152.22 | 144.90 | **9.73** |
| | Precision $\uparrow$ | 0.046 | 0.000 | 0.009 | 0.022 | **0.453** |
| | Recall $\uparrow$ | 0.000 | 0.000 | 0.000 | 0.000 | **0.614** |

Table 3: Results of multi-domain image generation on five datasets. We observe that the precisons of our method are slightly lower while the recalls are high across datasets, which is generally desirable since recall can be traded into precision via truncation, whereas the opposite is not true (Karras et al., 2020b; Kynkäänniemi et al., 2019).

There is a main implementation-related reason why we choose the first version: if we choose the second version, we have to wrap $m$ as a value inside [0,1]. A common way is to use sigmoid function. However, sigmoid is known for gradient vanishing (the gradient is very small when input is far from 0). So, our sparsity penalty and the GAN loss may have little effect on the sigmoid mask, which leaves the influence of domain very large. To testify the claim, we run experiments on AFHQ and CELEBA-HQ dataset with two versions in Table. 4. We set $\lambda = 0.1$.

| Mask Version | AFHQ | MNIST7 |
|---|---|---|
| | FID $\downarrow$ | FID $\downarrow$ |
| Version1: $z_s = \epsilon + m \odot f_u(\epsilon)$, | 4.36 | 1.42 |
| Version2: $z_s = (1 - m) \odot \epsilon + m \odot f_u(\epsilon)$ | 7.52 | 95.6 |

Table 4: The results of two mask mechanisms.

# I THE NECESSITY OF THE MASK MECHANISM

Our mask mechanism allows selecting the dimensions for injecting domain influences automatically. Another possible way seems to be manually define the dimension of style $n_s$ and tune it. It is worth noting that when one tries to find the optimal value of from 4,8,16,32.., in principle, one has to consider not only different values, but also which dimensions of the input of GAN should have changing distributions for each value of $n_s$, in order to achieve the best performance. Because of the complexity of the transformation implied by GAN, which dimensions of the input of GAN should have changing distributions may heavily depend on the initialization of GAN. That is, if we just allow the first inputs of GAN to have changing distributions, it is completely possible that other inputs, actually should learn to be one of those with changing distributions. In this case, it will be hard for GAN to learn the optimal function , and hence the strategy of forcing the first dimensions of the input of GAN to have changing distribution, even with the optimal value, may lead to high variability in the final performance. (On the other hand, it is not computationally feasible to consider each subset of the inputs of GAN of size , run the procedure, and find the best one.) This phenomenon is analogous to

| Datset | $n_s = 32$ | $n_s = 16$ | $n_s = 8$ | Ours |
|--------|-----------|-----------|----------|------|
| AFHQ   | 4.97      | 5.05      | 4.63     | 4.36 |
| MNIST7 | 2.0       | 3.7       | 32.6     | 4.87 |

Table 5: FID values when tuning $n_s$.

the relationship between traditional information criterion (like BIC)-based model selection or subset selection and parameter shrinkage (say, with the penalty) for variable selection.

## J  UNPAIRED IMAGE-TO-IMAGE TRANSLATION

### J.1  DETAILS ABOUT STARGAN-V2

As mentioned in section 3.3, we build our method based on StarGAN-V2 Choi et al. (2020). Now we provide more details about the method. StarGAN-v2 consists of four modules: mapping network $H(.,.)$, style encoder $E(.,.)$, the shared generator $F$ and the discriminator $D$. The training loss is

$$\mathcal{L}_{\text{stargan}} = \mathcal{L}_{adv} + \mathcal{L}_{\text{cyc}} + \mathcal{L}_{sty} - \lambda_{div}\mathcal{L}_{div}, \tag{35}$$

where $\mathcal{L}_{adv}$, $\mathcal{L}_{cyc}$, $\mathcal{L}_{sty}$ and $\mathcal{L}_{div}$ are used for *distribution matching*, *cycle consistency*, *style reconstruction* and *generation diversity*, respectively. StarGAN-V2 supports two kinds of tasks: latent-based image translation and reference-based image translation. Given an input image, latent-based generates the style code $\tilde{s}$ from the random noise $\epsilon$ with the mapping network $H$, i.e., $\tilde{s} = H(\epsilon, y)$, where $y$ is the target domain label. As for the reference based, the style is extracted from an image $x_2$ in the target domain $y$, i.e., $\tilde{s} = E(x_2, y)$. The shared generator takes an input image $x$ and the style code $\tilde{s}$ and outputs an image from the domain $y$. StarGAN-V2 concatenates multiple domain specific heads for each domain and we denote it as $D_y$ for domain $y$.

$$\mathcal{L}_{adv} = \mathbb{E}[\log D_y(x)] + \mathbb{E}[\log(1 - D_{\tilde{y}}(F(x, \tilde{s})))], \tag{36}$$

performs adversarial training between the generator and the discriminator.

$$\mathcal{L}_{sty} = \|\tilde{s} - \mathbb{E}(F(x, \tilde{s}))\|_1, \tag{37}$$

encourages the network $F$ contains the style information of $\tilde{s}$.

$$\mathcal{L}_{div} = \mathbb{E}[\|F(x, \tilde{s}_1) - F(x, \tilde{s}_2)\|_1], \tag{38}$$

which encourages two output images to be different by maximizing this loss.

The final one is the cycle consistency (Zhu et al., 2017),

$$\mathcal{L}_{cyc} = \mathbb{E}[\|x - F(F(x, \tilde{s}), \tilde{s})\|_1], \tag{39}$$

which encourages the network $F$ to reconstruct the input from the translated one. In other words, it encourages the mapping to be one-to-one.

### J.2  OUR TUPLE LOSS

However, as shown in previous literature (Alami Mejjati et al., 2018; Kim et al., 2019), the cycle consistency is not enough and can still leads to large content distortion. Hence, we introduce our loss

$$\mathcal{L}_{\text{tuple}} = \mathbb{E}\|F(G(\boldsymbol{\epsilon}, u_0), \tilde{s}_2) - G(\boldsymbol{\epsilon}, u_1)\|_1, \tag{40}$$

where $G(\epsilon, u_0), G(\epsilon, u_1)$ are pair data generated by our multi-domain image generation model. As proved in section F, the tuples can be viewed as sampling from the true joint distribution. Therefore, we are using $\mathcal{L}_{\text{tuple}}$ to encourage the mapping network $F$ to recover the second image from the first image. In other words, by minimizing this tuple loss, the network $F$ learns the correspondence relationship between our method. Therefore, our tuple loss helps further regularize the mapping network $F$ and avoid large content distortion.

### J.2.1 SETUP

**Implementation** We build on the official pytorch implementation of StarGAN-V2 (Choi et al., 2018). We set $\lambda_{\text{tuple}}$ to 0.1. We add our regularization loss in the every latent and reference iteration.

**Datasets and Metrics**. We use the benchmark AFHQ dataset (Choi et al., 2020) for image-to-image translation. It consists of three domains as stated before. Unlike the image generation task, we now have 6 pair-wise image-to-image translation tasks since we have 3 domains: cat→dog, cat→wild, dog→cat, dog→wild, wild→cat and wild→dog. We use the benchmark metrics: FID and LPIPS.

**Baselines**. We compare the results with MUNIT (Huang et al., 2018b), DRIT (Lee et al., 2018), MSGAN (Mao et al., 2019) and recent methods StarGAN-V2 (Choi et al., 2020), SmoothGAN (Liu et al., 2021), LETIT (Zhao & Chen, 2021) and StyleDis (Kim et al., 2022). We cite the results of MUNIT, DRIT, MSGAN and StarGAN-v2 from (Choi et al., 2020). StyleDis only reports the FID of reference based task. But it doesn't support the latent-based generation defined in StarGAN-v2.

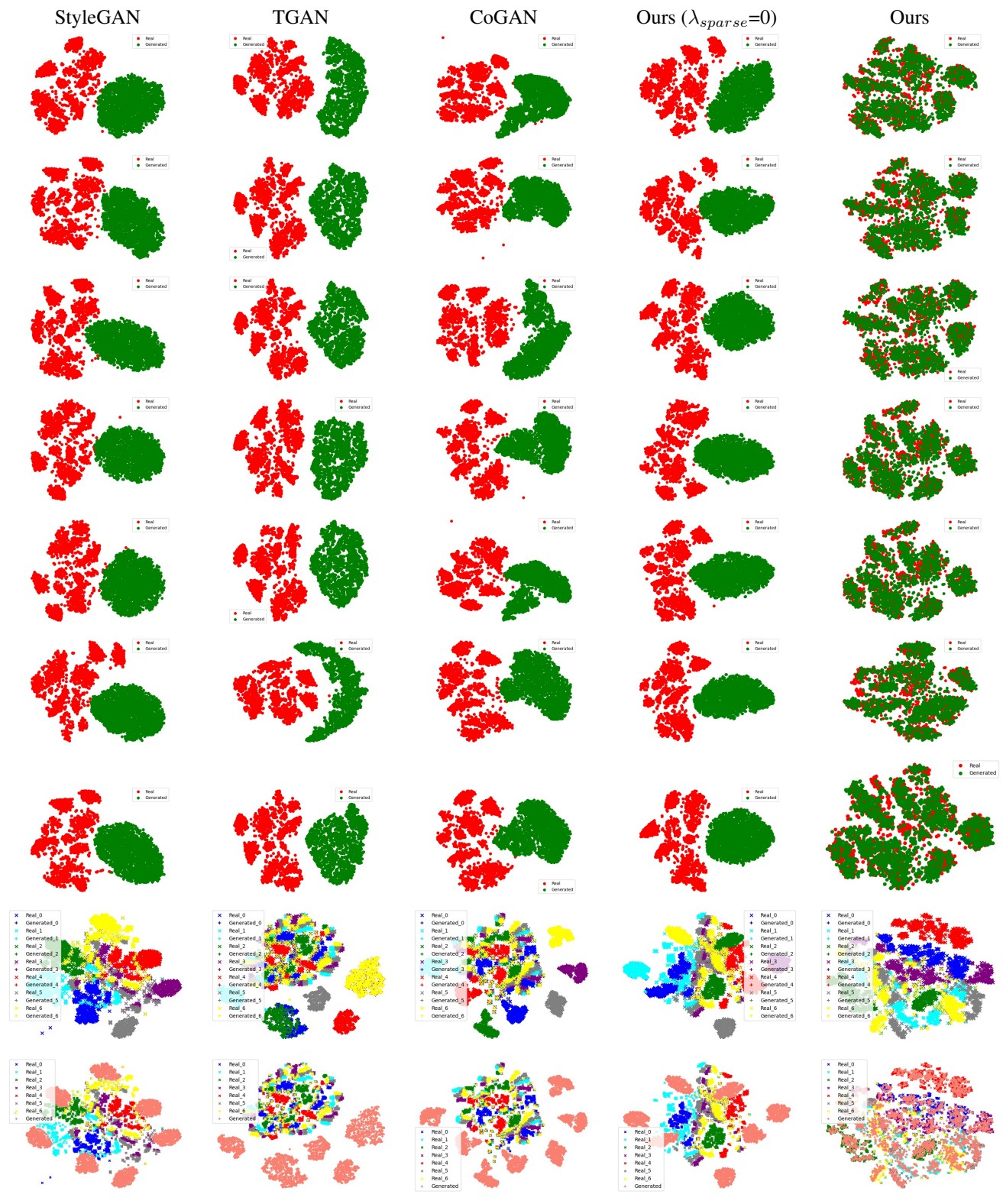

Figure 9: The t-SNE of real andgenerated samples for MNIST7 dataset. We can find that the baseline methods fail to recover the joint distribution. By contrast, our method matches the joint distributions.

StyleGAN2-ADA TGAN CoGAN $\lambda = 0$ $\lambda = 0.1$

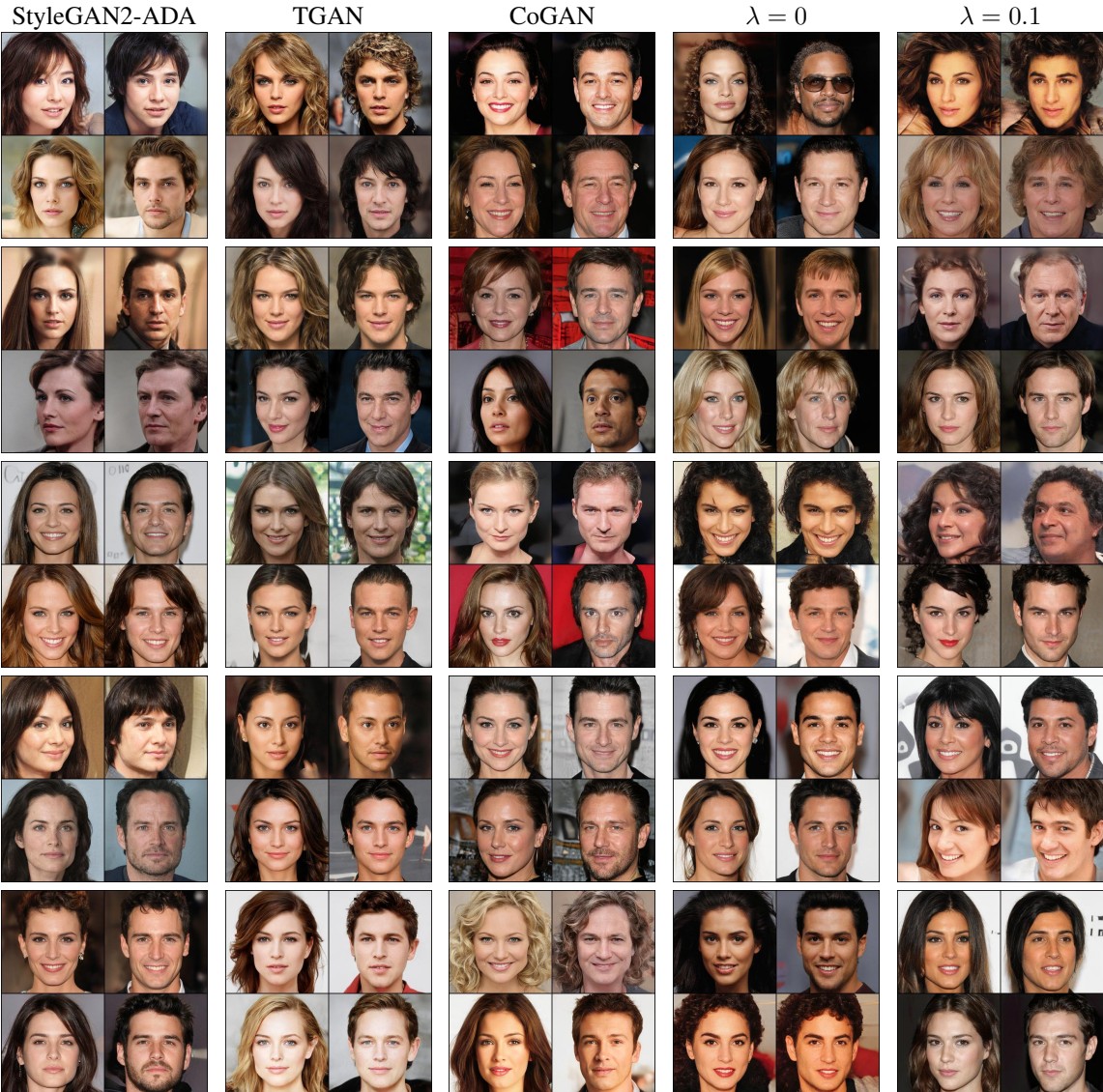

Figure 10: CelebA-HQ. Without the sparsity regularization, i.e., $\lambda = 0$, we observe some unnecessary changes between the image tuples in each row. For example, e.g.,the added sun-glasses and skin color change in the first row. TGAN changes the background (first row of third panel). CoGAN changes the skin color (second row, second panel).

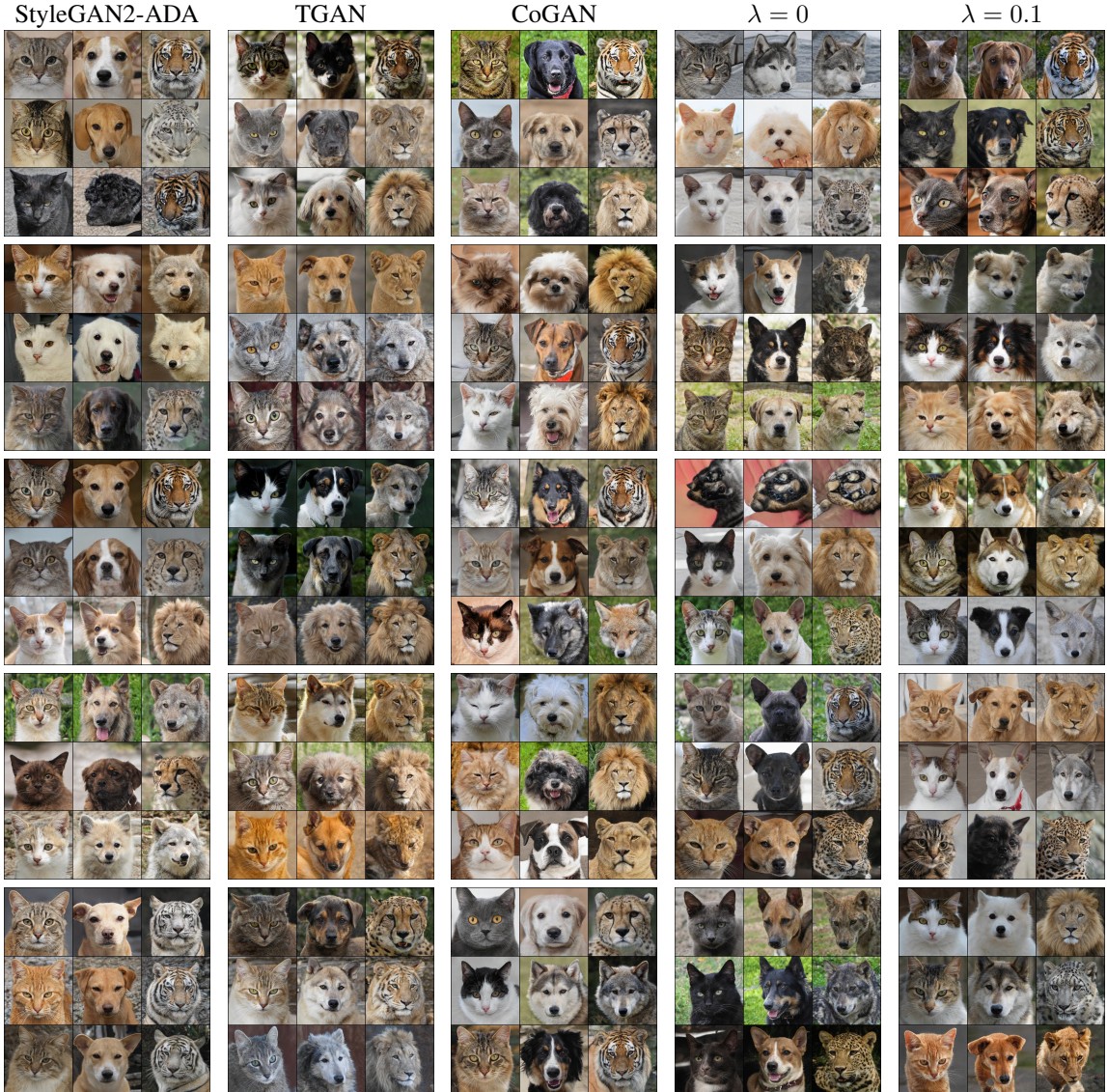

Figure 11: AFHQ. StyleGAN2-ADA changes animal poses in many examples, e.g., second and third row of first panel. Our base ($\lambda = 0$) also changes the poses, e.g., first and third row of second panel. CoGAN and TGAN are slightly better in preserving poses but we can observe that some generated images are unrealistic. For example, the wolf (first row, third panel of TGAN) and the dog (third row, third panel of CoGAN).

| StyleGAN2-ADA | TGAN | CoGAN | $\lambda = 0$ | $\lambda = 0.1$ |

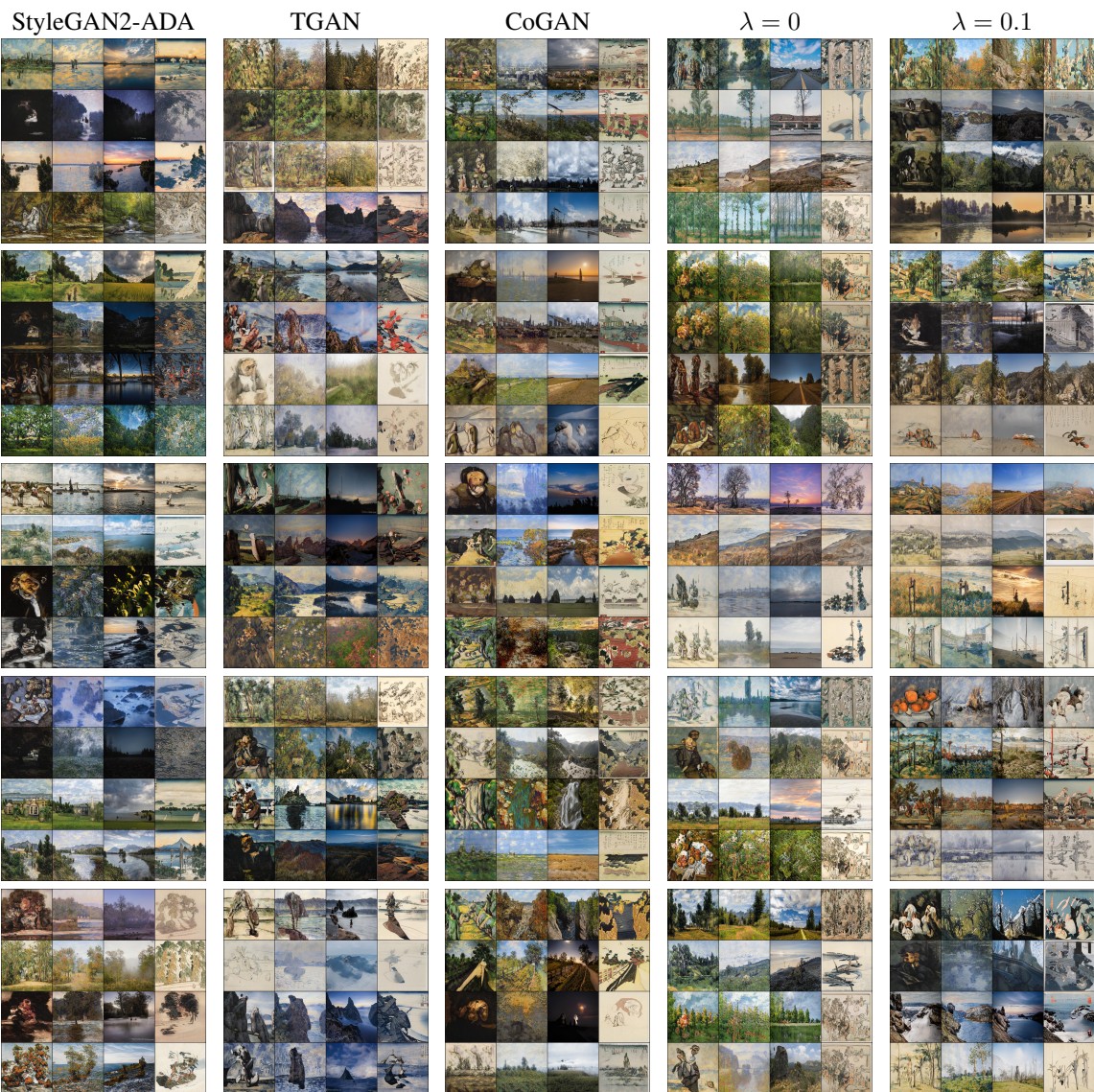

Figure 12: ArtPhoto. This case is probably the most challenging one since the content is not aligned. We can observe many changes between the images of the generated tuples of the baselines method, StyleGAN2-ADA, TGAN, CoGAN and our base $\lambda = 0$.

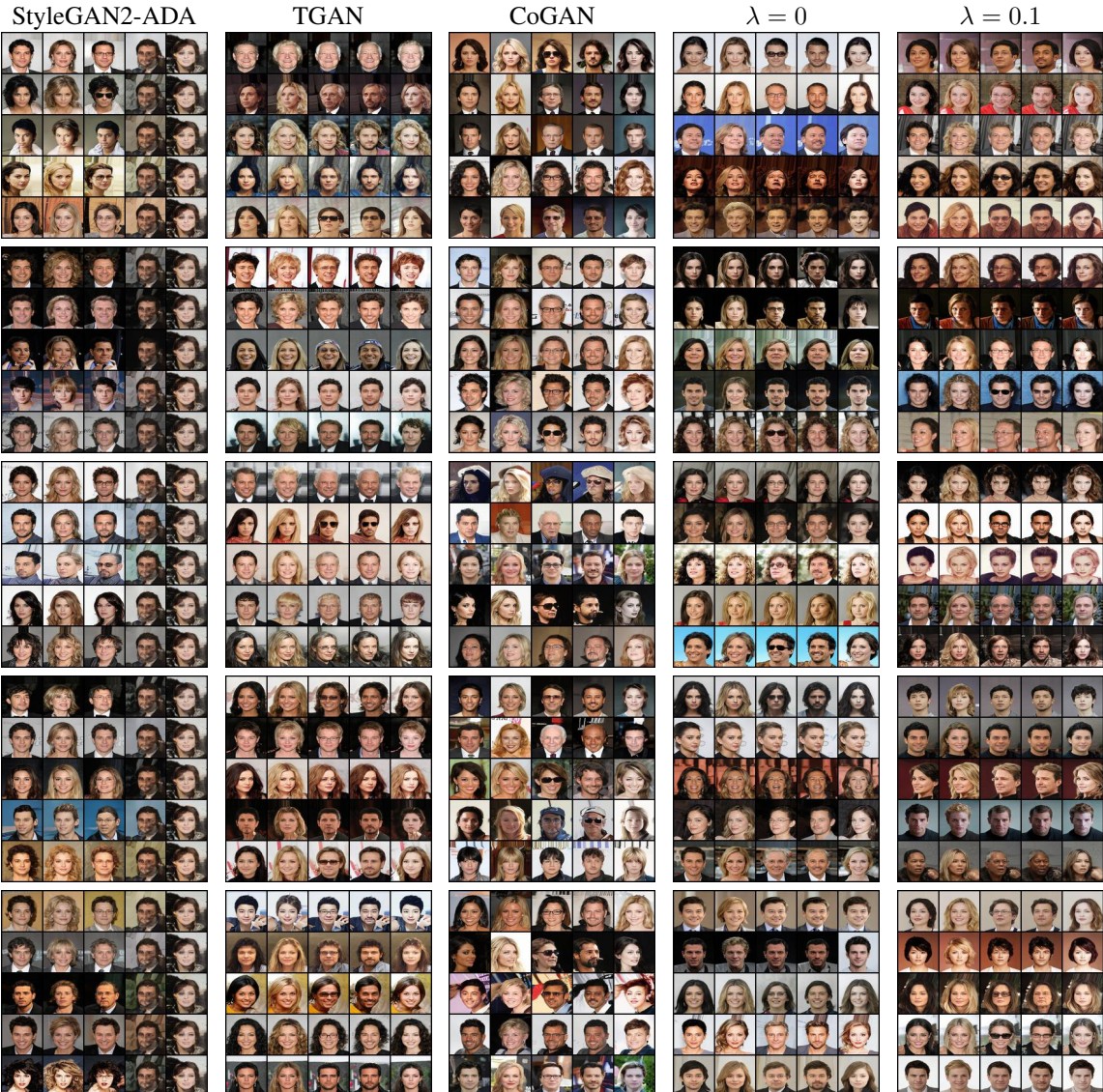

Figure 13: Celeba5. StyleGAN2-ADA suffers the mode collapse (samples in the columns are identical). TGAN and our base ($\lambda = 0$) often changes the background.

Figure 14: MNIST7. All baselines seem to suffer the mode collapse issue while TGAN seems to be affected by the augmentation in StyleGAN2-ADA negatively.

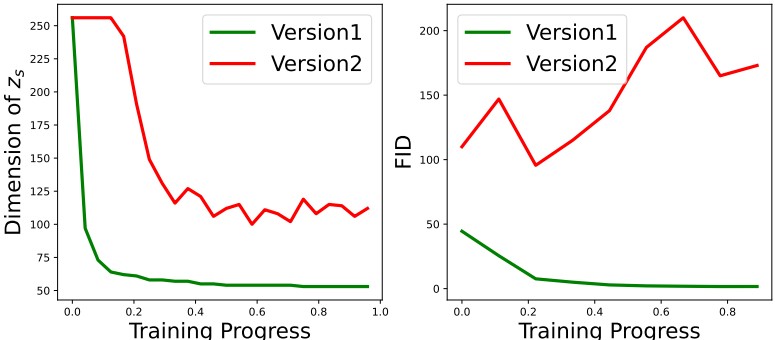

Figure 15: The trend of dimension of $z_s$ as training proceeds on MNIST7 dataset. We observe that the sigmoid version decreases much slower than our proposed version when we set $\lambda = 0.1$. Even if we set $\lambda = 1.0$, the speed of decreasing the dimension $z_s$ is smaller than our version. However, decreasing the dimension of $z_s$ is very important at the beginning phase as we need to reduce the influence of domain variable to avoid conditional collapse. The sigmoid version decreases faster than our proposed method in the middle of training progress. The reason is that the GAN already collapse and GAN loss is close to 0 and it helps little in selecting the optimal dimension $n_s$.

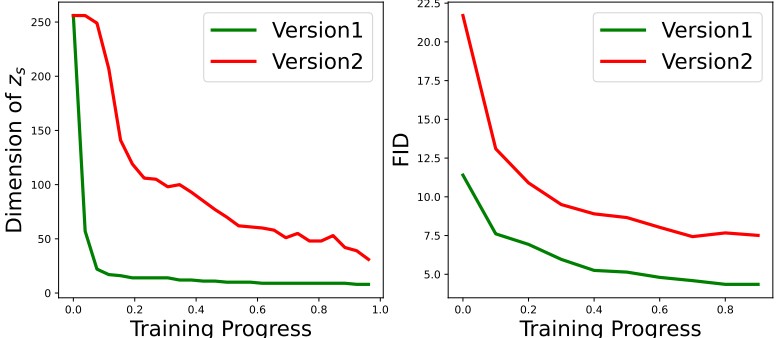

Figure 16: The trend of dimension of $z_s$ as training proceeds on AFHQ dataset. .

Input StarGAN-V2 Ours

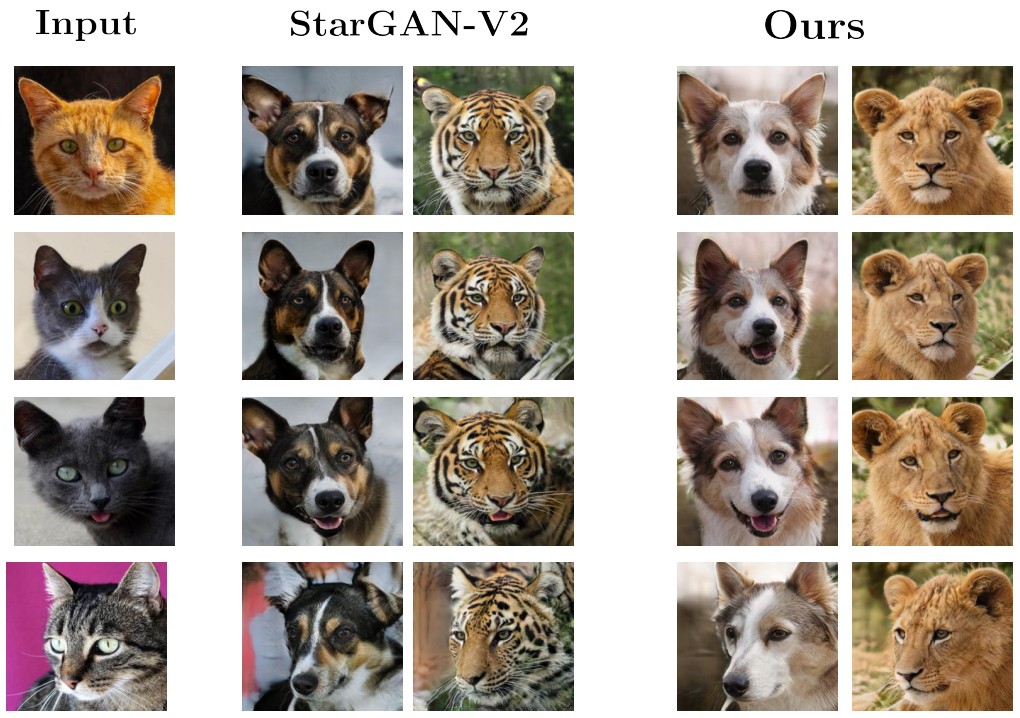

Figure 17: Latent-Based: cat→dog and cat→wild

Input StarGAN-V2 Ours

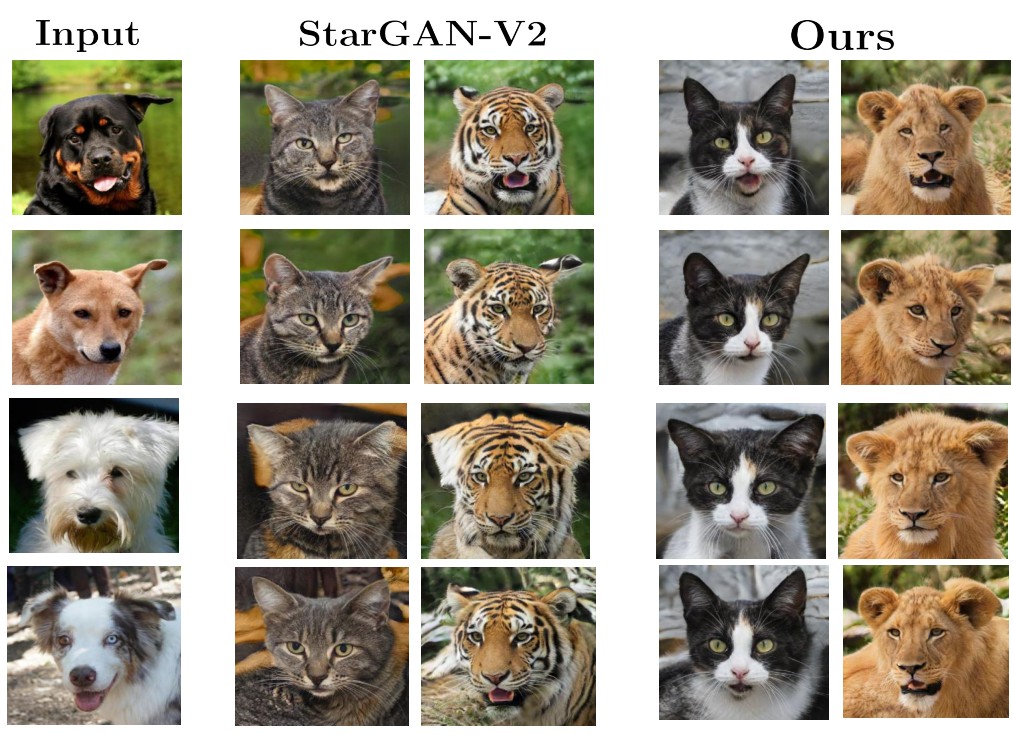

Figure 18: Latent-Based: dog→cat and dog→wild

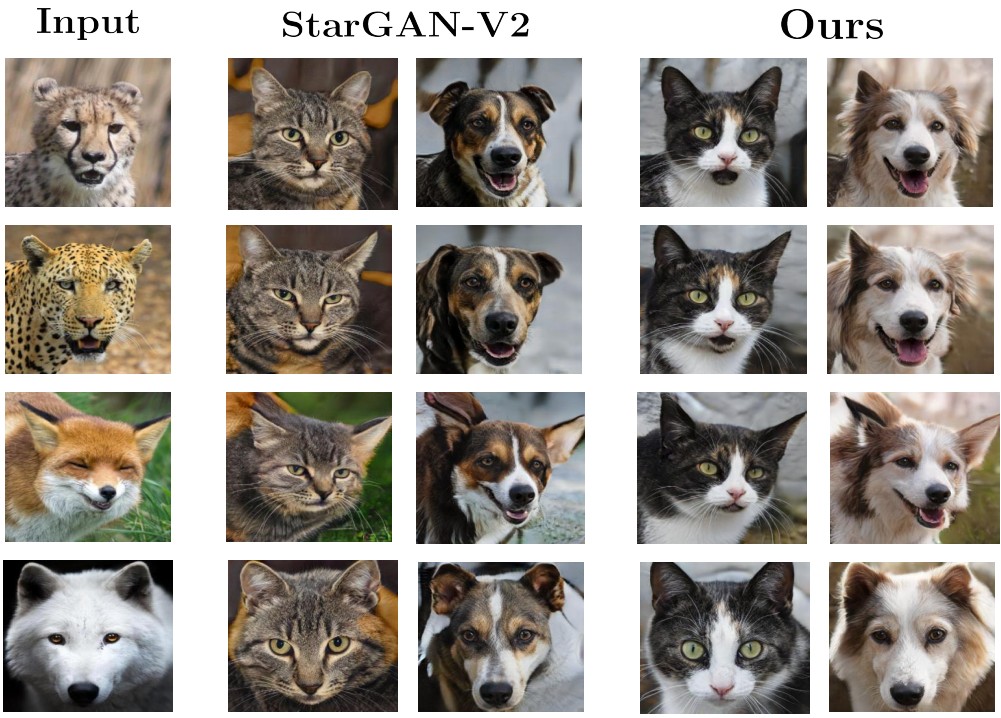

Figure 19: Latent-Based: wild→cat and wild→dog

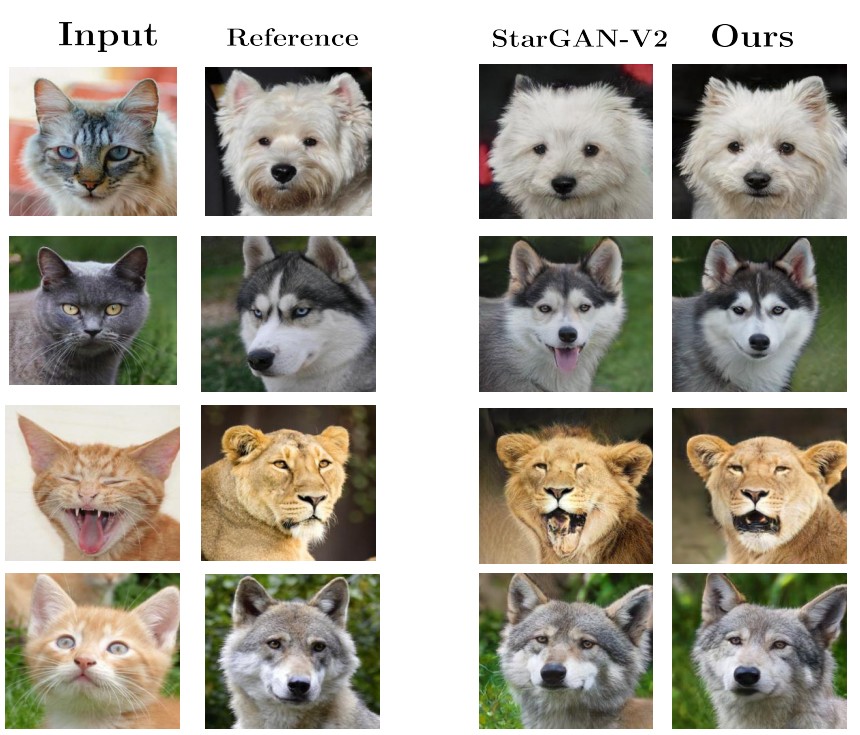

Figure 20: Reference-Based: cat→dog and cat→wild

**Input**    Reference    StarGAN-V2    **Ours**

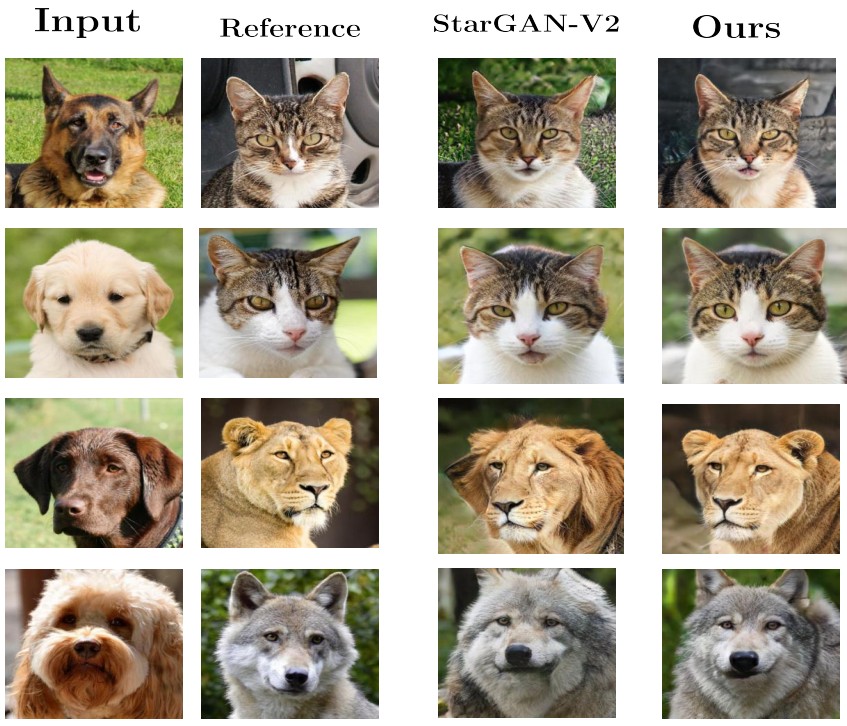

Figure 21: Reference-Based: dog→cat and dog→wild

**Input**    Reference    StarGAN-V2    **Ours**

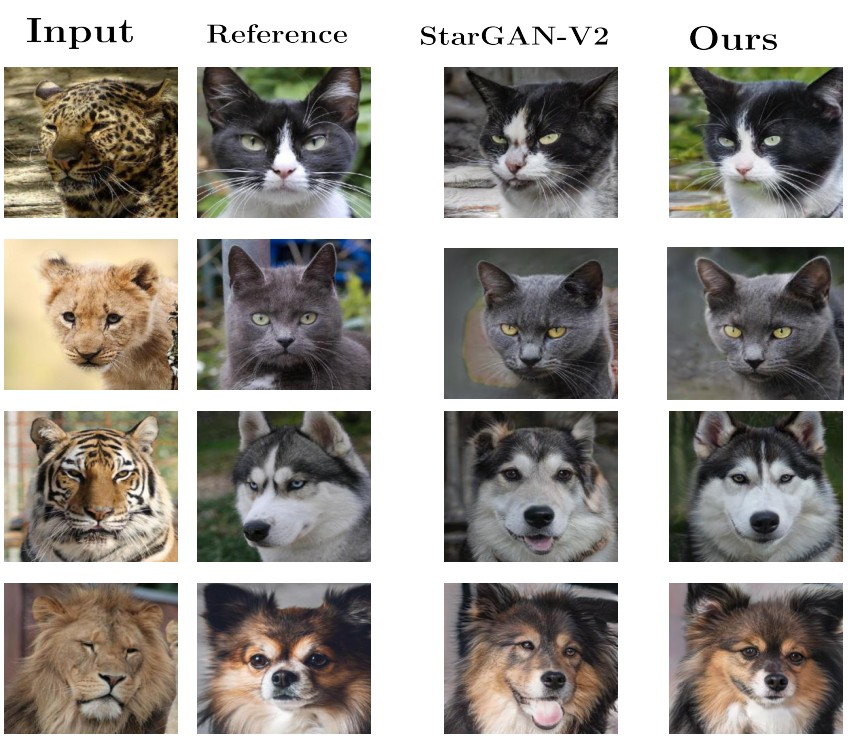

Figure 22: Reference-Based: wild→cat and wild→dog

