# OpenReview forum: "Multi-domain image generation and translation with identifiability guarantees"
_ICLR.cc/2023/Conference — ICLR 2023 notable top 25%_

### Official Review · Reviewer_1JgB · 2022-10-20

**Confidence:** 4
**Correctness:** 3
**Technical Novelty And Significance:** 3
**Empirical Novelty And Significance:** 4
**Recommendation:** 6

**Clarity, Quality, Novelty And Reproducibility:**

Clarity:
- What is \mathcal{Z} ? Undefined?
- Lemma 3.1:
    - Isn’t A1 a consequence of (4)?
    - A2 is also a consequence of (4). Indeed, if you apply an element-wise transformation to a vector of mutually independent random
    variables (which in this case is a Normal(0,I)), the resulting vector remains mutually independent.
    - In A2, what is said in words corresponds to pairwise conditional independence while what is written mathematically corresponds to mutual conditional independence, which are not the same.
    - A3. q is not defined.
    - A4. What are B_{z_c}, B_{z^*_c} ? What is Z_s? Both are undefined, thus I can’t understand this assumptions.
    - The conclusion of the theorem refers to “component-wise identifiability” and “block-wise identifiability”, but both terms were defined after the statement of the lemma, the author might want to consider introducing their definition before the lemma.
- Section 3.2
    - I do not understand the following statement: “If we feed the domain label into the generator directly, the generator may over-utilize the information of domain label u in order to confuse the conditional discriminator D”. What does over-utilize mean? What does it mean to confuse the conditional discriminator?
    - If we set n_s = n and n_c = 0, is it equivalent to feeding the domain label u in f_u ?
- Section 4.1
    - Caption of figure 3: “We observe that there are unnecessary changes between the images (e.g., the added sun-glasses in the first row, the different poses of animals of StyleGAN2-ADA in second row) without regularization.” I agree that this is indeed observed in the samples provided in Figure 3, but is it an observation that generalizes across samples?
    - The MNIST samples in Figure 3 for the baseline methods look a bit weak to me. Why is that? The baselines should give good sample quality, no? (Although we expect weaker pairing quality in the baselines)
    - I cannot really assess the relevance of the metrics used.

Novelty/originality:
- As far as I know, this is the first work proposing an identifiability result for this setting, which I think is a significant contribution.

Minor:
- The model p(z_s | u) defined in equation (4) factorises over the components of z_s, i.e. this is the usual conditional independence assumptions of the latents made in previous ICA work, this should be mentioned explicitly when introduced.
- In first paragraph of second page: set of marginals from 1 to n, but should it be from 1 to d?
- Equation (1) also has n instead of d.
- Extra parenthesis in (4)

**Strength And Weaknesses:**

Strengths:
- I very much like the overall direction of this work. I believe leveraging recent identifiability results from the nonlinear ICA literature for the problem of multi-domain image generation and image-to-image translation is a very good idea, given how important the question of identifiability is to these problems. Bringing this more theory-driven mindset to this literature should benefit the community.
- The experimental section appears fairly complete, to the eye of a non-expert in multi-domain image generation.

Weaknesses:
- Overall, some theoretical statements lack clarity. One finds many undefined notation which makes some of the statements impossible to understand. Also some mathematical statements lack precision. I expand on this in the "Clarity" section later in my review.
- The connections between some sections could be improved, especially between the theoretical results and the suggested algorithms. For example, in Section 3.2, I believe there is a mismatch between equation (6) and how the ground-truth model (4) was specified: in the model (6) z_s = \epsilon + f_u (\epsilon), whereas in the ground-truth model, z_s = f_u(\epsilon). This could be fixed in the model by doing z_s = (1 - m) * \epsilon + m * f_u(\epsilon). Is there a specific reason for not doing that?
- Section 3.3 is very hard to understand. A lot of new notation is introduced such as H, E, F and D without proper explanations. The function F is called the “shared generator” and then called the “mapping function”, this makes everything hard to follow. Also, it is not so clear to me exactly what approximates p_\theta(x^{(u_1)} | x^{(u_0)}). Overall, the connection with the theory from the previous section is not so clear to me.
- The motivation for using the masking mechanisms instead of just treating n_s as an hyperparameter is weak. An experiment attempts to show that the penalty coefficient associated to the mask is easier to tune than n_s. But I found this experiment unconvincing, since a priori, it’s unclear what the range of \lambda we should search over… The left top plot of Figure 4 shows a flat curve, but the curve would not look so flat had the range been different. How does one choose the proper range? All main experiments in the paper are performed with \lambda = 0.1 and yields very good results. What happens when these experiments are run without this masking mechanism and with n_s = 16 or 32 (which were the best dimensionality found in the ablation study)? Do we see similar performance?
- I couldn’t find information in the main text about how the mask m is learned, nor a reference to appendix. Is m = sigmoid(parameter) ? Or is it a Gumbel-Sigmoid?




**Summary Of The Paper:**

This paper leverages recent theoretical developments in nonlinear ICA to introduce a novel model for multi-domain image generation and show that its joint distribution is identifiable from the marginal distributions. Motivated by this result, two practical algorithms are suggested, one for multi-domain image generation, which corresponds to sampling from the joint over domains, and one for image-to-image translation, which corresponds to sampling from a conditional. Both of these approaches are based on the assumptions that only a few latent variables are influenced by the domain label, which is implemented in the learned model via learned masks and regularization. Extensive experiments are performed, comparing multiple baselines on multiple datasets, which all point in the direction of improved matching between samples from the joint and better image-to-image generation.

**Summary Of The Review:**

Even though I very much like the overall direction of this paper, (i) the overall lack of clarity, (ii) the lack of cohesion between the theory and the proposed methods and (iii) the lack of evidence for the usefulness of the masking mechanism prevent me from recommending acceptance. I am open to increase my  score if the authors address the main points I raised in my review (see Weaknesses section).

---

> ### Author Response · Authors · 2022-11-18
> **Response to Reviewer  1JgB**
>
> Thank you so much for the valuable comments and suggestions. We are glad that you like our work and we also hope that our paper could benefit the community.  We have revised the paper and included new results according to your suggestion. We hope the revision could address your concerns.
>
> ## The clarity.
>
> **Missing definitions of notations, i.e., $\mathcal{Z}$, $q$, $B_{z_c}$, $B_{z_s}$, $Z_s, Z_c$.**
>
> Thank you for the careful comments. We have added the definitions of these notations in the revised version (above lemma 3.1; highlighted in red).
>
> **The assumptions**
>  A1 is a specific condition on the probability density function. It is not necessarily the consequence of equation 4. However, if we assume that P(z_s) and P(z_c) are normal distributions, and that P(z|u) is smooth (the second-order derivative of the log density exists), then A1 holds true. We also explained in what sense the density is smooth.
>
> As for A2, We are so grateful for your careful comment. Indeed, pairwise independence and mutual independence are not the same, and we used mutual independence in the proof. We have revised the A2 assumption accordingly. Yes, equation (4), together with the setting that $\tilde{\text{z}}_s, \text{z}_c$ are independent gaussian variables, implies that A2 is satisfied.
>
> As for A3 and A4, intuitively, they require that the changes of latent variables are sufficiently rich across domains such that we have sufficient contrastive information to achieve the identifiability.  A3 requires that the changes in the probability density functions are complex enough such that they do not always lie within (2n_s+1)-dimensional subspace.
>
>
> ## Method
>
> **the generator may over-utilize the information of domain label u**
>
> To generate images in each domain, the generator needs to utilize the input domain label $\textbf{u}$. However,  the influence of $\textbf{u}$ can be very large if no constraint is applied.  If the influence of domain variable is unnecessarily large, the generator focuses on the domain variable u and pays little attention to the content variable $\textbf{z}_c$. As a consequence, the generated images in the tuple may lose correspondence (e.g., the generated animals have different poses on AFHQ dataset, Fig. 4.). An extreme case would be that the content is totally ignored by the generator. So the generator still outputs images in different domains (because the input domain labels are different ) but all images in the same domain look the same (because the content variable $z_c$ is ignored) (e.g., the generated digits look the same on MNIST7 dataset, Fig. 4).
>
> When we mean “confuse the discriminator”, we refer to the adversarial training between the generator and the discriminator. The discriminator is trained to distinguish the generated samples from real samples in each domain and the generator is trained to confuse the discriminator (to increase the loss of the discriminator). We removed this term in the revised submission to avoid potential confusion. Thanks!
>
> **If we set $n_s = n$ and $n_c = 0$, is it equivalent to feeding the domain label u in $f_u$ ?**
>
>  We feed domain label u to construct function $f_u$ regardless the value of $n_s, n_c$.  If we set $n_s=n, n_c=0$, we are adding domain influence on all dimensions of the latent variable and the large influence may leads to the missing of correpondence across domains, which is unwanted for multi-domain image generator and image-to-image translation.
>
> **Whethter the observation generalizes across samples**
>
> We provided more generated samples in the appendix.  We also use the metric DIPD to measure whether the generated images share more content than other baseline methods. The lower values of DIPD across datasets demonstrate that generated images by our method share more content than other methods.
>
> **The MNIST samples in Figure 3 for the baseline methods look a bit weak**
>
> The main reason why baseline methods fail is that they suffer the conditional collapse at the very beginning due to the unconstrained influence of domain variable. Then the GAN losses become meaningless and the generator stops improving the quality of generated samples. When there are only two domains, StyleGAN-ADA avoids the collapse issue and generates realistic samples although the digits have different labels (bottom panel of Fig. 5(b)).

---

> > ### Comment · Reviewer_1JgB · 2022-11-22
> > **Response**
> >
> > **The mask mechanism:**
> >
> > I'm actually not sure this equivalence is valid, since the scaling depends on a learnable parameter, i.e. the mask itself. It's like saying y = x**2 is equivalent up to rescaling to y = x because y/x = x. No?
> >
> > It is also worth noting that this "equivalence" holds only when m_2 is different from 1 (otherwise there is a division by zero). At first I thought this could be a problem because I thought m_2 was in [0,1], but I believe it is in [0, \infty), right? It might be worth being more explicit in the paper about how exactly this mask is learned and what is his domain.
> >
> > **Motivation to use mask mechanism:**
> >
> > > We added experiments on AFHQ and the best result is achieved when n_s = 8  (table 5 in appendix)
> >
> > > For example, with , our method learns to select 50 for MNIST7 dataset and 9 for AFHQ dataset (which are close to the optimal dimensions by sweeping the n_s).
> >
> > I think these empirical observations are much more convincing then the current argument from the manuscript, please add them to the paper.
> >
> > > If we use the mask mechanism, the mask is trained with the sparsity loss and the GAN loss. If the learned dimension is too small, the GAN loss will be high and encourage the model to preserve more dimensions to be . If the learned dimension is too large, the sparsity loss will be high and encourage the model to drop some dimensions to be . So, it learns to find the optimal dimension for each dataset automatically.
> >
> > This is a bit hand-wavy, but ok there is some empirical support.
> >
> > > Of course, \lambda = 0.1  may not always be the optimal value but the search range of \lambda is significantly smaller than the search range of n_s.
> >
> > I find this argument unconvincing. How can we know [0,1] is easier to search then {4, 8, 16, 32, ...} ?

---

> > > ### Author Response · Authors · 2022-11-25
> > > **Response to Reviewer 1JgB**
> > >
> > > We are grateful for your prompt response, in light of which, we conducted experiments to verify our theoretical claim.  (Sorry for keeping you waiting; we were doing experiments in the last two days.)
> > >
> > > **The equivalence of two mask mechanisms**
> > >
> > > Thanks for your question. The equivalence is valid because the scaling is a parameter and is not related to the input noise $\epsilon$. In your example,  we agree that the claim that $y=x^2$ is equivalent to $y=x$ up to rescaling may not make sense since $x$ is the changing variable.  But, in our case, the main changing variable is the Gaussian noise $\epsilon$ rather than the mask.  After training, the masks are fixed and we generate different samples from different input Gaussian noise $\epsilon$. If we have $m=\frac{m_2}{1-m_2}$, the outputs of the two versions only differ in scale for the same input Gaussian noise $\epsilon$. These two versions are two different parameterizations and we choose the first version owing to practical reasons. They are like the two different implementations of VAE: we may use the output of the encoder as standard deviation directly. But that may cause some implementation issues (e.g., we have to enforce the outputs to be non-negative since deviation is non-negative). So, a more popular way is to use the output of the encoder as log variance and further transform it into standard deviation.
> > >
> > > Yes, when m_2 is exactly 1, there will be a division by 0. But that would be a very rare case in practice since m_2 is a continuous variable. If our m is big enough, it can approximate the case when m_2 is close to 1. Yes, the domain of our mask m is $[0, +\infty)$.
> > >
> > > Thanks for your suggestion. We have added the domain of the mask and the empirical observations to the main paper of our manuscript.
> > > We also have added a discussion about these two mask mechanisms in our manuscript. We believe that the paper is significantly improved with your valuable suggestions.
> > >
> > > **Motivation to use mask mechanism**
> > >
> > > Thanks for your constructive feedback. We have included more experiments to demonstrate the necessity of using the mask mechanism.
> > > First of all, we completely agree that [0,1] may be a larger range than {4,8,16,32..}.  At the same time, it is worth noting that when one tries to find the optimal value of $n_s$ from {4,8,16,32..}, in principle, one has to consider not only different $n_s$ values, but also which dimensions of the input of GAN should have changing distributions for each value of n_s, in order to achieve the best performance.  Because of the complexity of the transformation implied by GAN, which dimensions of the input of GAN should have changing distributions may heavily depend on the initialization of GAN.
> > > That is, if we just allow the first $n_s$ inputs of GAN to have changing distributions, it is completely possible that other inputs, such as the $(n_s+1)$th dimension, actually should learn to be one of those with changing distributions. In this case, it will be hard for GAN to learn the optimal function $g$, and hence the strategy of forcing the first $n_s$ dimensions of the input of GAN to have changing distribution, even with the optimal $n_s$ value, may lead to high variability in the final performance. (On the other hand, it is not computationally feasible to consider each subset of the inputs of GAN of size $n_s$, run the procedure, and find the best one.)
> > >
> > > Our new experiments verified this claim. The table below shows the FID of two mechanisms on MNIST7 dataset across 7 random runs. You can see that the result with $L_1$ penalty together with $\lambda = 0.1$ is good and stable. However, although $n_s = 32$ may produce good results in some runs, its FID value has a high standard deviation (12.9, in contrast with 1.5 for the method with $L_1$ penalty).
> > >
> > > This phenomenon is analogous to the relationship between traditional information criterion (like BIC)-based model selection or subset selection and parameter shrinkage (say, with the $L_1$ penalty) for variable selection [1].  We would appreciate it if you let us know what you think of this point.
> > >
> > > [1] Tibshirani, Robert. "Regression shrinkage and selection via the lasso." Journal of the Royal Statistical Society: Series B (Methodological) 58.1 (1996): 267-288.
> > >
> > > | Mechanism      | Run1  | Run2 | Run3 | Run4 | Run 5| Run 6 |Run7| Mean| STD|
> > > | ----------- | ----------- |----------- |----------- |----------- |----------- |----------- |----------- |----------- |----------- |
> > > | $n_s$=32      | 2.0|21.5| 24.4|3.0|2.8| 35.8| 2.0|13.1 | 12.9|
> > > | $\lambda=0.1$   | 1.4|3.0|1.8|1.6|2.8|2.0|6.1|2.7|1.5|

---

> > > > ### Comment · Reviewer_1JgB · 2022-12-02
> > > > **Response**
> > > >
> > > > I thank the authors for their response.
> > > >
> > > > I still disagree about the equivalence of the two masking mechanisms. I found the explanation unconvincing. The parameters m_1 and m_2 varies, just like x in y = x^2, so I don't understand why the authors insist on thinking of it as a constant.
> > > >
> > > > I find the hypothesis about why the masking mechanism is useful with GANs compelling. And there seems to be empirical support for it as well. All in all, I believe the case for the masking mechanism is much stronger in light of the new evidence. Also, my complaints about clarity have been mainly addressed. I'm thus happy to raise my score to 6 and recommend acceptance. I encourage the authors to include these arguments/experiments/clarification about the masking mechanism in the paper.

---

> > > > > ### Author Response · Authors · 2022-12-03
> > > > > **Thank you so much for your engagement and response and updating the recommendation.**
> > > > >
> > > > > We are really happy that most of your concerns were properly addressed.  We also agree with you that the mask mechanisms differ in some ways (thanks for it) and will not claim they are equivalent. We definitely will follow your valuable suggestions and include the arguments/experiments/clarification about the masking mechanism in the paper, which are essential for the improvement of the paper.
> > > > >
> > > > > Thanks again for your devoted time and constructive feedbacks!

---

> ### Author Response · Authors · 2022-11-18
> **Response to Reviewer 1JgB (2)**
>
> **the relevance of the metrics used**
>
> The main reason why we use the metrics is that we don’t have samples from ground truth joint distribution (except the MNIST dataset). So, on MNIST dataset, we measure the divergence between the joint distributions of generated samples and the ground truth joint distribution (JointFID). On other datasets, we first use FID to measure the divergence between marginal distributions of generated samples and real samples.  Then we use DIPD to measure how much the content is shared across domains.  We use FID and DIPD as a surrogate of the divergence between the joint distributions of generated samples and the ground truth joint distribution. We also included the precision and recall metrics in the appendix.
>
> **The mask mechanism**
>
> Thanks for your comments.  The first version: $z_s=\epsilon+m\odot f_u(\epsilon)$ and  the second version  $z_s^{(2)} = (1-m_2)\epsilon + m_2\odot f_u(\epsilon)$ are actually equivalent up to scale transformation. $z_s^{(2)}=(1-m_2)\odot \epsilon + m_2\odot f_u(\epsilon)$ can be reformulated as $\frac{1}{1-m_2}z_s^{(2)}=\epsilon + \frac{m_2}{1-m_2}f_u(\epsilon)$. For example,  $z_s^{(2)}=0.5\epsilon + 0.5 f_u(\epsilon)$ when $m_2=0.5$. In our case, if $m=1$, we have $z_s=\epsilon+f_u(\epsilon)=2 z_s^{(2)}$.  There is a practical reason why we choose the first version: if we choose the second version, we have to wrap $m$ as a value inside [0,1]. A common way is to use $sigmoid(parameter)$. However, sigmoid is known for gradient vanishing (the gradient is very small when input is far from 0). So, our sparsity penalty and the GAN loss may have little effect on the sigmoid mask during training, which leaves the influence of domain very large.  To testify the claim, we run experiments on AFHQ and MNIST7 dataset with two versions and we present the results in Table. 3 and Fig. 15,16 in the appendix.  The large influence of domain variables at the beginning of the training causes the mode collapse and FID becomes very bad.  The second version might work after careful design and tuning. But the first version is more stable according to current results.
>
> **notations in Section 3.3.**
>
> Thanks for pointing it out. We corrected the terms and add figure 3 to illustrate the proposed regularization. The mapping function $F$ aims to translate the images in one domain to another domain. So, we use our regularization to encourage $F$ approximate the true conditional distribution $p_{\theta}(x^{(u_1)}|x^{(u_0)})$, which is learned by our multi-domain image generation model $G(\epsilon, \theta)$.
>
> **motivation to use mask mechanism**
>
> Thanks for your input.  In the main paper, we show that the result is best when $n_s=32$ on MNIST7 dataset and the results are bad when $n_s$ is smaller than 16. The problem of treating $n_s$ as hyper-parameter is that the optimal value highly depends on the dataset, so 32 may not be an optimal value for other datasets.  We added experiments on AFHQ and the best result is achieved when $n_s=8$ (table 5 in appendix). Therefore, we need to sweep at least [8,16,32] for each dataset if we use $n_s$ as hyper-parameter. If some datasets are very complex, we may also need to consider $n_s=64$ or larger.
>
> If we use the mask mechanism, the mask is trained with the sparsity loss and the GAN loss. If the learned dimension is too small, the GAN loss will be high and encourage the model to preserve more dimensions to be $z_s$. If the learned dimension is too large, the sparsity loss will be high and encourage the model to drop some dimensions to be $z_s$. So, it learns to find the optimal dimension for each dataset automatically.  For example, with $\lambda=0.1$, our method learns to select 50 for MNIST7 dataset and 9 for AFHQ dataset (which are close to the optimal dimensions by sweeping the $n_s$).  As we shown in the main paper, $\lambda=$0.1 works well across all datasets and we think [0,1] is a reasonable range for such $\lambda$. Of course, $\lambda=0.1$ may not always be the optimal value but the search range of $\lambda$ is signficantly smaller than the search range of $n_s$.
>
>
> According to the training time cost in https://github.com/NVlabs/stylegan2-ada-pytorch, we can save at leat 44(hours) $\times$4(GPUs)$\times$(3-1)(tasks)=352 GPU hours for each dataset (we use 4 GPUs to train the model).  If we apply our model on 1024x1024 resolution datasets (thanks to the scalability of the code of StyleGAN), we can save at least 290$\times$4$\times$(3-1)=2320 GPU hours for each dataset.
>
> **how the mask m is learned**
>
> We use $parameter$ as the mask directly. To avoid negative values, we clip it to 0 if value is less than 0.1.  We didn’t use sigmoid due to the gradient vanishing problem.  We provide the code in the supplementary material: *Line 229, Line 263-268 in code_i-stylegan/training/networks.py*.
> Your second proposal: Gumel-Sigmoid sounds like a good alternative and we will try it.

---

### Official Review · Reviewer_spME · 2022-10-24

**Confidence:** 4
**Correctness:** 3
**Technical Novelty And Significance:** 2
**Empirical Novelty And Significance:** 4
**Recommendation:** 6

**Clarity, Quality, Novelty And Reproducibility:**

I have a few concerns about this paper.

1) The motivation to use ICA is not really clear. Is it to build the true joint distribution? Also I am wondering whether the diversity are reduces with ICA regulation? Could authors show the precision/recall?, such as StyleGAN and the proposed method, and starganv2 and the proposed one?

2) As reported in Table 1, on mnist dataset the baseline (stylegan) has 95.9 FID, while the proposed method is 1.43. I am wondering how author perform the comparison experiment.

3) The proposed method is easily influenced by hyper-parameters.

4)  In this introduction, authors mentions current methods fail to remove the unwanted joint distributions, which means this paper can achieve it. could authors give some examples? For example, the age or does not change when changing hair color. I fail to find some kinds of figures like this.

**Strength And Weaknesses:**

Although there are a few outstanding methods proposed to explore conditional image generation, this paper explores a new view, which is effective to remove unwanted joint distributions.

This looks solid which give strong theory support about how to guarantee the true joint distribution.

The paper is well-written, and easy to follow.

Authors provides  conditional image generation and image-to-image translation results, which sounds interesting.

**Summary Of The Paper:**

This paper focuses on conditional image generation and image-to-image translation. Although current methods enable both tasks work well, they fail  to formulate suitable constraints for the joint distribution, since there can be infinitely many joint distributions that can derive the same marginals. Inspired by Independent Component Analysis (ICA) theory, authors propose a new regularization by enforcing a specific type of minimal changes across domains. The quantitative and qualitative results demonstrate the effectiveness of the proposed method.

**Summary Of The Review:**

This paper sounds interesting, which first considers ICA into the training of both conditional image generation and image-to-image translation.  The main experiment  shows the effectiveness of the proposed method.

---

> ### Author Response · Authors · 2022-11-18
> **Response to Reviewer spME**
>
> Thank you so much for the insightful comments and suggestions. We believe that ICA is a promising direction to address the multi-domain image generation and translation problems.
>
> **The motivation of ICA**
>
> Yes, ICA is used to build the true joint distribution. In our setting, we are only given observations (images) from different domains and we don’t have access to the correspondence information (e.g., the class label of digits, the identity of the human). We aim to generate paired data (e.g., sharing class label or identity) from unpaired data (e.g., digit images). In order to generate meaningful pairs from unpaired data, we need to recover the true content and style latent variables (and thus recovering the true joint distribution). Nonlinear ICA is famous for recovering the (unknown) latent variables (content and style) from observations (e.g.,images).
>
> **Including Precision and Recall**
>
> Thanks for your suggestion. Due to space limitations,  we included the precision/recall in Table. 3 in the appendix. We observe that our method achieves lower precision while higher recall compared to baseline methods. As pointed out by the StyleGAN-V2 paper, this is generally desirable since recall can be traded into precision via truncation, whereas the opposite is not true. An example would be the unconditional FFHQ generation tasks by StyleGAN2 and StyleGAN. StyleGAN2 improves the Recall from 0.399 to 0.492 while the precision drops from 0.721 to 0.689 (table 1 in the StyleGAN2 paper)
>
> **Evaluation of FID**
>
> We use the metrics (https://github.com/NVlabs/stylegan2-ada-pytorch/blob/main/calc_metrics.py ) provided in the code of StyleGAN2-ADA. We train the baseline models with their recommended hyper-parameters. Specifically, we generate 50K samples and compare them against all real images.  The reason why the FIDs of baseline methods are bad is that they suffer from the well-known conditional mode collapse problem (as shown in Fig. 4).  Through minimizing the influence of domain variable, our method is able to avoid such failure and generate realistic digit samples.  We also provide the t-SNE results of all methods in the appendix F.3 (Fig. 9).
>
> **The proposed method is easily influenced by hyper-parameters**
>
>  If we set $n_s$ as hyper-parameter, then the results are easily influenced by the value of $n_s$ (as shown in top right, Fig. 5 (a)).  To address this issue, we propose a mask mechanism to encourage the networks to select the optimal value of $n_s$ automatically (section 3.2). So, our hyper-parameter is $\lambda$ instead. We found that 0.1 works well across all datasets.  We also have tested the robustness of $\lambda$ by changing it to different values on MNIST7 dataset. We can find that the method works well if $\lambda$ is within the range [0,1] (top left, Fig5 (a))), which is a common range of hyper-parameters.
>
> **Example of recovering joint distribution**
>
> Sure, we showed some samples in Fig 4. In particular, the MNIST7 samples can demonstrate our claim. Please note that we are only given digit images from 7 colors of MNIST and we don’t have access to the class label of the digits, so we don’t know the correspondence across domains. The samples of the true joint distribution should share the same class label while showing different colors. Existing methods fail to generate such tuples. By contrast, our method is able to generate digits with the same class label but different colors. This proves that the joint distribution is recovered. We also compute the FID between the joint distribution of real and generated digits on MNIST7 datasets (JointFID). The low value of JointFID demonstrates that the recovered joint distribution is very close to the ground truth joint distribution.

---

> > ### Comment · Reviewer_spME · 2022-11-29
> > **Keep my score**
> >
> > Thanks for rebuttal. Reading authors'comment, I would like to keep score, and positive to this paper, which in fact has contribution to the community.

---

> > > ### Author Response · Authors · 2022-11-29
> > > **We are very grateful for your time, valuable comments, and positive feedback.**
> > >
> > > Thank you so much for your time, valuable comments, and positive feedback. We are really encouraged by your appreciation of our work.
> > >  And your suggestions definitely made our paper better!

---

### Official Review · Reviewer_h5sT · 2022-10-24

**Confidence:** 5
**Correctness:** 3
**Technical Novelty And Significance:** 3
**Empirical Novelty And Significance:** 3
**Recommendation:** 8

**Clarity, Quality, Novelty And Reproducibility:**

#### **Clarity**
- The paper is well written and easy to follow.
- The key motivation and overall idea toward addressing the task is clearly explained.
- The implementation way is simple and reasonable.
- The code is well organized.

#### **Quality**
- The model reaches better performance than the state-of-the-art methods.
- The effectiveness of the theory and model is clearly demonstrated in the experiments.

#### **Novelty**
- The cheap normalization is novel and interesting.
- The implementation for the join distribution through a simple domain-specific component-wise monotonic transformation is reasonable.

#### **Reproducibility**
- The code is well organized. I believe the experimental results can be reproduced using the provided code.

**Strength And Weaknesses:**

#### **Strengths**
- There are many works using the StyleGAN or StarGAN architectures to perform the image generation and translation tasks but without in-depth analyses. This paper provides an in-depth study of the joint distribution for multi-domain generation and translation, which provdes some insights for the following researchers.
- The implementation way is easily achieved, and the loss function is interesting. The corresponding impressive results are good enough to support the contribution claimed by the authors.
- The paper is well organized and is easy to follow. Several observations are studied with quantitative and visual analyses, followed by the corresponding ablation study, making the paper easy to follow and concrete.

#### **Weaknesses**
While the proposed method is interesting and the experimental analyses are comprehensive, I believe some parts need more clarification (even after considering the supplementary material and code).
- The joint distribution is the key motivation and novelty in the paper. However, the experiments just demonstrate the much better performance and the effectiveness of the provided framework. It will be better to visualize the distribution of the original dataset as well the generated results of different methods through the t-SNE, especially for MINIST7, which contains obvious domain and class category.
-  The domain $u$ is predefined. As claimed by the authors, "collecting corresponding data across domains can be prohibitively expensive". Then, how could the authors define the domain numbers in a dataset?
- More generally, for the CELEBA-HQ, the authors split it as female and male faces domains. Could we also add other domains, such as sad and happy, young and old and others? Does different domain numbers affect the performance?

**Summary Of The Paper:**

This paper provides an in-depth study of estimating the joint distribution of multi-domain image generation and unpaired image-to-image translation. To mitigate the highly ill-posed issues of mapping from marginal distributions to joint distribution, the authors provide an in-depth analysis for the distribution modeling, while their theory is based on some tight assumptions. Besides, the paper implements this theory in a cheap way. Specifically, the paper empirically demonstrates the disentangled content and style controllable generator through a simple framework and loss function. Then, based on this architecture and new regularization, the paper improves the performance of multi-domain image generation and unpaired image-to-image translation on multiple datasets.

**Summary Of The Review:**

This paper investigates an important topic of unified multi-domain image generation and unpaired image-to-image translation task. Despite the experimental part could be further enriched to clearly demonstrated the better joint distribution learned by the proposed method, the easily achieved implement and thorough analyses on the properties provides many insights and might arouse following researchers. Therefore, my rating for this submission is positive.

---

> ### Author Response · Authors · 2022-11-18
> **Reponse to Reviewer h5sT**
>
> Thank you so much for the valuable comments and suggestions. We also hope that our method could provide insights for the following research. We have revised and included some results and we hope the revision could address your concerns.
>
> **t-SNE visualization of the generated images and real images**
>
> Thanks for your suggestion. We totally agree that t-SNE would help illustrate the effectiveness of our method. We have included the t-SNE of real and generated images for MNIST7 dataset in Fig. 5 (c). We also included the t-SNE comparisons of all domains of MNIST7 dataset in the appendix F.3 (figure 9).  The t-SNE visualizations further support our theory and the method.
>
> **the domain number of u**
>
> Thanks for the question. We meant that paired data are difficult to collect while the unpaired data can be collected easily. We are not trying to partition the dataset into different domains automatically. Still, we need to collect multi-domain data without correpondence. For example, when we aim to generate female and male pairs, we just collect two domains: female images and male images. So, in our setting, the number of domains is 2.
> The question you raised may inspire an interesting future direction!  We understand it as how to partition the data into different domains when they are mixed to produce a single large dataset. Please kindly let us know if we misunderstood you. We believe it is a very important but challenging problem and hope to contribute to it in the future.
>
> **can we add other domains?**
>
> Sure, we can add other domains to the dataset. In fact, we also conducted experiments on the CelebA5 dataset, which contains 5 domains: Black Hair, Blonde Hair, Eyeglasses, Mustache, and Pale Skin (this dataset is similar to the dataset you suggested). We can find that our method achieves the best FID and DIPD on this dataset (table 1). We believe we can obtain similar results when the domain changes to sad, happy, young and old and others. (By the way, the CELEA-HQ dataset contains male and female face domains, not split by us.)
>
> **different domain numbers affect the performance?**
>
> Indeed, different domain numbers affect the performance.  As shown in bottom right of Fig. 5 (a) and section 4.1.2, we decreased the domain numbers in MNIST7 from 7 to 2 gradually, and observed that the performance is better when there are more domains. This aligns with our theory (A3 in lemma) (we need 2$n_s$+1 domains for the identifiability of the latent content and style).  It is worth mentioning that even if we only have 2 domains, our method still achieves satisfactory results (bottom panel of Fig. 5 (b) and section 4.1.2).

---

### Official Review · Reviewer_6m6y · 2022-10-29

**Confidence:** 3
**Clarity, Quality, Novelty And Reproducibility:** See above
**Correctness:** 4
**Technical Novelty And Significance:** 3
**Empirical Novelty And Significance:** 3
**Recommendation:** 6

**Strength And Weaknesses:**

Strengths:
1. The proposed approach shows pretty good improvement over the baseline methods, especially on the task of image-to-image translation.

2. The ablation studies show the effect of different design choices, such as reducing the dimension of the style dimension or the effect of the number of style dimensions on generation quality. These are informative in understanding the effect of design choices on generation quality.

Weaknesses/questions:
1. The quality of writing in the paper is uneven, with some parts not written clearly, which makes the paper difficult to understand in parts.

2. As the authors also allude to in the last section, the proposed method is not as effective when the content across domains is not aligned, such as the case of the ArtPhoto dataset (where the method shows the least improvement in FID). Can the authors comment the general applicability of the method in the light of this limitation, beyond standard constrained facial image and digit datasets?


**Summary Of The Paper:**

This method proposes a method for multi-domain image generation and unpaired image-to-image translation. The main challenge in these problems is learning a joint distribution from multiple marginal distributions is ill-posed, since there can be infinitely many joint distributions that can derive the same marginals.  Utilizing recent advances in nonlinear Independent Component Analysis (ICA) theory, the authors propose a new method to learn the joint distribution from the marginals. With the assumption that the influence of domain information is minimal in the data generation process, the method introduces the domain information through a component-wise monotonic transformation. Further, by assuming that the number of the underlying components affected by the domain information is minimal, the authors show that the true joint distribution can be recovered from the marginal distributions. Then the learned joint distribution can be used to sample meaningful tuples and translate input images to another domain without content distortion. First, the authors test the method on five multi-domain image generation datasets and evaluate using FID and domain-invariant perceptual distance (DIPD) metrics (for datasets with unpaired data.) On this dataset, the method obtains some improvement over baseline approaches, and especially shows good results on MNIST7, where other methods fail due to mode collapse. On the application of image-to-image translation, the method also obtains good results, outperforming baseline methods in both latent-based and reference-based settings.


**Summary Of The Review:**

The paper proposes a method for multi-domain image generation and unpaired image-to-image translation, using an ICA-based method to learn the joint distribution from the marginals. The proposed method obtains decent improvement over baselines on standard datasets, however, performance gains are limited on complex images where the content across domains may not be aligned.

---

> ### Author Response · Authors · 2022-11-18
> **Response to Reviewer 6m6y**
>
> Thank you so much for your valuable comments.
>
> **The quality of writing in the paper is uneven**
> Thanks for your comment, which helped improve the presentation. We have revised the theory in Section 3.1, rephrased the discussion in Section 3.2, and added an illustration for Section 3.4.  We hope this revision improves the readability of the paper. If you find the writing still uneven, please kindly let us know and we will address it immediately.
>
> **The general applicability of our method**
>
> Thank you for your insightful suggestion. We have included the applicability of our method in Section 1 (highlighted in red). Our method can be applied to datasets where the content across domains are aligned and most existing datasets in multi-domain image generation and translation satisfy this requirement, e.g., the facial image dataset, animal face and digit images. Our method may not be as effective when the contents when some domains contain unaligned contents. We may need to pay more attention to the artworks-related dataset, since different painters focus on different objects in their painting (e.g., there are many fruit paintings by Cezanne and landscape paintings by Monet).

---

> > ### Comment · Reviewer_6m6y · 2022-12-07
> > **Response to rebuttal**
> >
> > Thanks for your response. I am satisfied with your responses to both of my questions and I am happy to keep my score.

---

> > > ### Author Response · Authors · 2022-12-08
> > > **Thank you so much for your time and positive feedback**
> > >
> > > We are really glad that our responses have addressed your questions! Thank you again for your time and constructive comments!

---

### Decision · Program_Chairs · 2023-01-20

**Decision:**

Accept: notable-top-25%

**Justification For Why Not Higher Score:**

After the discussion phase, all reviewers advocated acceptance, highlighting the novelty of the theory and the strong empirical results. This paper is unusual in how it connects a popular area of applicatoin -- unpaired image generation and translation -- to theoretical results on identifiability. The AC therefore feels it deserves a spotlight as it may be of broad interest to multiple communities. However, as it does not break entirely new ground on either applications or theory, the AC feels it falls short of the level of an oral.

**Justification For Why Not Lower Score:**

See above.

**Metareview: Summary, Strengths And Weaknesses:**

Summary:
This paper addresses multidomain image generation and unpaired image-to-image translation, treating them as two instances of estimating a joint distribution from marginals. Using recent theory from ICA, the paper shows certain conditions under which the joint distribution is identifiable, then implements a model that satisfies these conditions (by making some assumptions about the distributions). The results are strong on both multidomain generation and translation.

Strengths:
* One of the first results connecting unpaired image generation and translation to identifiability.
* Strong empirical results

Weaknesses:
* Clarity could be improved, especially regarding the connection between the theory and the practical method

**Note From Pc:**

if the above contains the word "oral" or "spotlight" please see: "oral" presentation means -> notable-top-5% and "spotlight" means -> notable-top-25%. As stated in our emails, we are disassociating presentation type from AC recommendations